# An intrinsically disordered region mediates RNA-binding selectivity and cellular activities of LARP6

Federica Capraro [1,2,11], Giancarlo Abis [2,11], Alessio Incocciati [2,3], Peter J. Simpson [4], Mehran Karimzadeh [5], Laura Masino [6], Alexander Barley [7], Tam T. T. Bui [8], Geoff Kelly[4], Hani Goodarzi [5,9], Maria R. Conte [2] ✉ & Faraz K. Mardakheh [1,10] ✉

Intrinsically disordered regions (IDRs) are prevalent in RNA-binding proteins (RBPs), yet their roles in RNA interactions remain poorly defined. We examined RNA-binding regulation by structured and disordered regions of LARP6, an RBP with a diverse RNA-binding repertoire. Mass spectrometry-based RNA interaction mapping in living cells identified direct LARP6–RNA contacts within the structured La-module and its flanking IDRs. Mutagenesis and individual-nucleotide resolution UV-crosslinking and immunoprecipitation (iCLIP) revealed the La-module, but not the IDRs, as essential for LARP6 RNA binding. Deletion of the N-terminal IDR broadened LARP6 RNA footprints, uncovering a role in RNA-binding selectivity. This is achieved through a composite mechanism of restricting the conformational flexibility of the adjacent La-module, forming auxiliary contacts with the RNA, and modulating RNA access for binding. The IDR-mediated RNA-binding selectivity is critical for LARP6-mediated promotion of cancer cell viability and invasion. Our findings uncover a previously unrecognised critical function for IDRs in promoting selective RBP–RNA recognition, by affecting the binding specificity of their adjacent structured domains.

RNA-binding proteins (RBPs) control all aspects of RNA regulation and function[1]. Conventional RBPs have been traditionally defined by the presence of one or more structurally conserved RNA-binding domains (RBDs), which mediate interactions with their target RNA molecules through sequence- or structure-based RNA recognition[2]. The human genome encodes hundreds of such RBPs, which can be grouped into diverse super-families, characterised by the presence of specific sets of RBDs[3]. Through combinatorial action of multiple RBDs, RBPs are thought to engage with and regulate specific sets of transcripts in context-specific manners, creating highly diverse yet specific post-transcriptional regulatory networks[4,5]. In addition to containing well-defined RBDs, most conventional RBPs also harbour intrinsically disordered regions (IDRs) that are characterised by an ensemble of interconverting conformations. Despite lacking defined structure, IDRs can play critical functions in various RBPs[2]. Indeed, the majority of germ-line mutations in RBPs that are associated with human

[1]Centre for Cancer Cell & Molecular Biology, Barts Cancer Institute, Queen Mary University of London, London, UK. [2]Randall Centre for Cell and Molecular Biophysics, King's College London, London, UK. [3]Department of Biochemical Sciences "Alessandro Rossi Fanelli", Sapienza University of Rome, Rome, Italy. [4]MRC Biomedical NMR Centre, The Francis Crick Institute, 1 Midland Road, London, UK. [5]University of California, San Francisco, CA, USA. [6]The Francis Crick Institute, 1 Midland Road, London, UK. [7]Independent researcher, London, UK. [8]Pharmaceutical Sciences, King's College London, London, UK. [9]Arc Institute, Palo Alto, CA, USA. [10]Department of Biochemistry, University of Oxford, South Parks Road, Oxford, UK. [11]These authors contributed equally: Federica Capraro, Giancarlo Abis. ✉e-mail: sasi.conte@kcl.ac.uk; faraz.mardakheh@bioch.ox.ac.uk

Mendelian genetic disorders occur within the IDR-coding regions[1]. Interestingly, unbiased RNA–protein interaction site-mapping studies have revealed that many IDRs with diverse biochemical characteristics can directly engage with RNA in living cells[6,7]. However, the exact mechanisms through which these 'non-conventional' disordered RNA binding sites interact with RNA, and how they combine with tethered RBDs to shape the RNA-binding properties of RBPs remain largely uncharacterised.

The La-related protein 6 (LARP6) is an evolutionarily conserved RBP found in all multicellular eukaryotes. LARP6 is a member of the LARP superfamily, a group of ancient eukaryotic RBPs involved in various aspects of post-transcriptional gene regulation, including the synthesis, processing, maturation, localisation, stability, and translation of diverse RNA targets[8]. Similar to all LARPs, LARP6 is characterised by the presence of a conserved bipartite RBD, the La-module, which consists of a La-motif and an RNA Recognition Motif-1 (RRM1), connected by a short linker[9]. This La-module is N- and C-terminally flanked by IDRs. In addition, LARP6 possesses a SUZ-C motif in its C-terminal region, a structurally uncharacterised short motif that is known to mediate protein–protein interactions[8]. Functionally, LARP6 has been shown to control diverse cellular processes, including type-I collagen synthesis[9,10], myogenesis[11–13], oocyte development[14], neural tube development[15], as well as pollen tube guidance in plants[16]. Recently, we identified a role for LARP6 in regulating mRNA localisation, cell proliferation, and invasion, in highly invasive breast carcinoma cells that have undergone epithelial–mesenchymal transition (EMT)[17]. Transcriptome-wide iCLIP analysis revealed over five thousand mRNAs, spanning diverse functional gene categories, as direct binding targets of LARP6, highlighting a capability to engage with a broad spectrum of RNA targets[17]. Consistent with these results, an RNA-based high-throughput systematic evolution of ligands by exponential enrichment (HTR-SELEX) study demonstrated that LARP6 has a highly complex RNA-binding repertoire, capable of interacting with various linear and structured motifs[18]. The underlying mechanism by which LARP6 can selectively recognise such a diverse array of RNA targets remains unclear.

Here, we devised an unbiased mass spectrometry approach to map the regions of LARP6 that directly interact with RNA in living cells. Our results reveal that, in addition to the structured La-module, regions within the N- and C-terminal IDRs of LARP6 can directly contact the RNA. Using a doxycycline-inducible system to express full-length (FL) or various deletion mutants of LARP6, coupled with transcriptome-wide iCLIP analysis, we demonstrated that the La-module is essential for the RNA-binding activity of LARP6 in living cells. Removing the IDRs did not abrogate LARP6's ability to interact with RNA. Instead, deletion of the N-terminal IDR broadened LARP6's interaction footprints across the transcriptome, indicating that this IDR modulates the activity of the adjacent La-module by promoting selectivity in RNA binding. In vitro biophysical assessments corroborated our observations in living cells, revealing that the N-terminal IDR enhances RNA-binding selectivity via a composite mechanism that restricts the conformational dynamics of the La-module, forms additional direct contacts with the RNA, and modulates RNA access to the La-module in a competitive manner. Importantly, deletion of the N-terminal IDR compromised the function of LARP6 in promoting the viability and invasiveness of glioblastoma cells, suggesting that the IDR-mediated RNA-binding selectivity is important for LARP6 function. Together, our findings uncover a previously unknown yet critical mechanism for IDRs in promoting selective RNA–RBP interactions of their adjacent structured domains.

## Results
### LARP6 interacts with RNA through both ordered and disordered regions
To better understand the complexities of LARP6 RNA-binding activity, we aimed to define its regions of contact with RNA in living cells in an unbiased manner. For this purpose, we developed a mass spectrometry-based approach termed IP-OOPS (Immunoprecipitation coupled with Orthogonal Organic Phase Separation) (Fig. 1a). This approach combines the isolation of UV-C crosslinked RNA–protein adducts via OOPS[19], with the ability to map protein regions that are attached to RNA by sequential Arg-C and trypsin digestions, as pioneered in the RBDmap method[6]. Briefly, GFP-trap beads were used to purify LARP6–RNA complexes from UV-crosslinked MDA-MB231 cells that stably expressed GFP-LARP6[17] (Supplementary Fig. 1a). The RNA-crosslinked LARP6 was then isolated using an initial OOPS, before partial digestion with Arg-C, which generated smaller segments, with or without RNA adducts. The RNA-bound segments were further purified from the unbound segments via a second OOPS, followed by trypsin digestion, releasing MS-detectable peptides that were adjacent to a crosslinked region (Fig. 1a). The crosslinked peptides themselves are undetectable in this approach, due to the addition of RNA mass. However, through identification of their adjacent peptides, the likely sites of crosslinking can be postulated as either N- or C-terminal tryptic regions flanking the identified peptides.

Using IP-OOPS, we identified three RNA-bound regions within LARP6 (Fig. 1b, c). These included a known RNA-binding area within the La-motif, as well as two previously unknown RNA-binding areas in the N- and C-terminal regions of the protein (Fig. 1b, c). As control, analysis of the non-crosslinked LARP6 from the organic phase revealed 21 peptides that were distributed across the full length of protein, and covering over 51% of the LARP6 sequence (Supplementary Fig. 1b), suggesting good coverage and a near uniform observability of different protein regions in our experiments. The C-terminal flanking candidate RNA-binding residues in the La-motif contains phenylalanine 135, one of six highly conserved residues in the RNA-binding pocket of LARP6, and shown by mutagenesis to be essential for RNA association[9], providing further confidence in our methodology. The RNA-binding area in the C-terminal region (CTR) of the protein is adjacent to the SUZ-C motif, while the RNA-binding area in the N-terminal region (NTR) precedes the La-module (Fig. 1b, c). Computational prediction of disorder using IUPred2A[20] suggested that both the NTR and CTR are likely intrinsically disordered (Fig. 1d). These results reveal that in addition to the folded La-module, LARP6 contains two previously unknown RNA-binding regions in its N- and C-terminal parts, which are likely to be IDRs.

### iCLIP analysis reveals the La-module as essential for LARP6 binding to RNA
Next, we set out to study the RNA targets of each RNA-binding region of LARP6 in living cells. To reveal the full repertoire of LARP6 binding targets, we chose U-87 MG glioblastoma cells as our cell model, since LARP6 is highly expressed in gliomas (Supplementary Fig. 2a), with U-87 MG cells being among the highest LARP6-expressing established cell lines in wide use (Supplementary Fig. 2b). In fact, LARP6 protein expression levels in U-87 MG cells even surpass those of MDA-MB231 cells, which were previously used as a cell model for LARP6 function, making them a suitable cell model for investigating LARP6 function, particularly in the context of cancer (Supplementary Fig. 2c). To assess the contribution of each region of LARP6 to its RNA-binding activity, we used a doxycycline-inducible system to express FL and various deletion mutants of a myc-tagged LARP6, in a controlled manner (Supplementary Fig. 2d–g). These cells were then subjected to iCLIP analysis using beads conjugated to a monoclonal anti-myc tag antibody[21] (Fig. 2a). The iCLIP experiment was performed in triplicate for each LARP6 variant, with parental U-87 MG cells serving as a negative control.

The resulting iCLIP reads, representing precise RNA–protein crosslink sites, were subsequently mapped to the human transcriptome. We then searched for clusters of reads across the transcriptome, which identified peaks that correspond to the likely LARP6

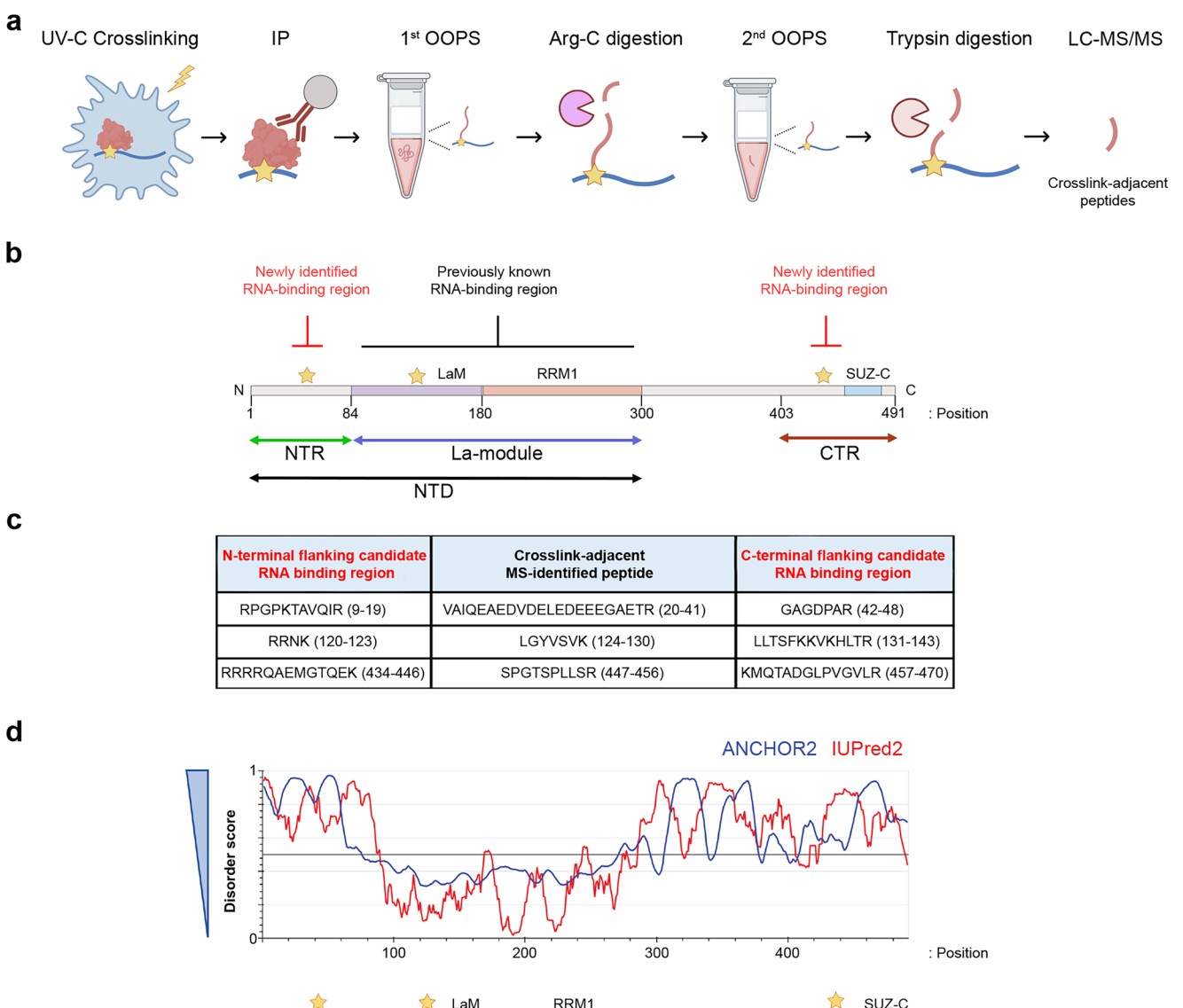

**Fig. 1 | LARP6 harbours two disordered RNA-binding regions. a** IP-OOPS workflow (Created with modifications using BioRender. Mardakheh, F. (2026) https://BioRender.com/16v6db4). The detailed protocol is described in Material and Methods and in the text. Star: crosslinking site; red: protein; blue: RNA. **b** Schematic representation of the RNA-binding regions identified in LARP6 via IP-OOPS-identified RNA-binding regions are marked by stars. LARP6 domains and regions are represented to scale, with numbers denoting the indicated residues at the start and end of each indicated region (LaM: La-motif; RRM1: RNA Recognition Motif 1; SUZ-C: SUZ domain-containing protein 1 C-terminal). The LaM and the RRM1 comprise the La-module, which together with the N-terminal region (NTR) constitute the N-terminal Domain (NTD). The SUZ-C and its adjacent N-terminal segment comprise the C-terminal region (CTR). **c** IP-OOPS-identified peptides and their N- or C-terminal flanking candidate RNA-crosslinked regions. The identified crosslink-adjacent peptides are retained in the interface of the 1st OOPS due to their proximity to the RNA crosslink sites but are released in the 2nd OOPS from the crosslink sites following trypsin digestion and detected by mass spectrometry. **d** Computational disorder prediction for human LARP6 protein sequence. The web server IUPred2A (https://iupred2a.elte.hu/)[20,65] was used to predict the likelihood of disorder for LARP6, based on ANCHOR2 or IUPred2 algorithms. Higher disorder scores denote higher likelihoods of the protein region being intrinsically disordered. The IP-OOPS-identified RNA-binding regions are marked by stars.

binding sites (Supplementary Data 1). A total of 14,334 peaks were detected for the FL LARP6, most of which were lost upon deletion of the La-module (Fig. 2b), demonstrating its crucial role for the RNA-binding activity of LARP6. In contrast, deletion of the NTR or CTR did not reduce the number of identified peaks; instead, the number of mapped peaks were increased in the ΔNTR and ΔCTR mutants (Fig. 2b, Supplementary Data 2, & Supplementary Data 3). Amongst the identified peaks that were conserved across FL, ΔNTR, and ΔCTR mutants, we detected the well-known binding sites of LARP6 in the 5′UTR of Collagen-I mRNA[10], and the 5′TOP sequences of various ribosomal protein mRNAs (RP-mRNAs)[17] (Fig. 2c). In agreement with our previous study[17], mapping of the peaks to the different genomic regions revealed the vast majority to correspond to mRNAs, with the highest binding density occurring in the 3′UTRs, a binding pattern that was largely unaffected by the NTR or CTR deletions (Fig. 2d).

We then assessed the functional categories of mRNAs that associate with LARP6. Annotation enrichment analysis of LARP6 targets revealed an overrepresentation of transcripts involved in ribosome biogenesis, cell adhesion and migration, intracellular trafficking, and RNA processing, amongst others (Fig. 2e). Importantly, this target repertoire was also unaffected by the NTR or CTR deletions (Fig. 2e). Together, these findings suggest that unlike the La-module, NTR and

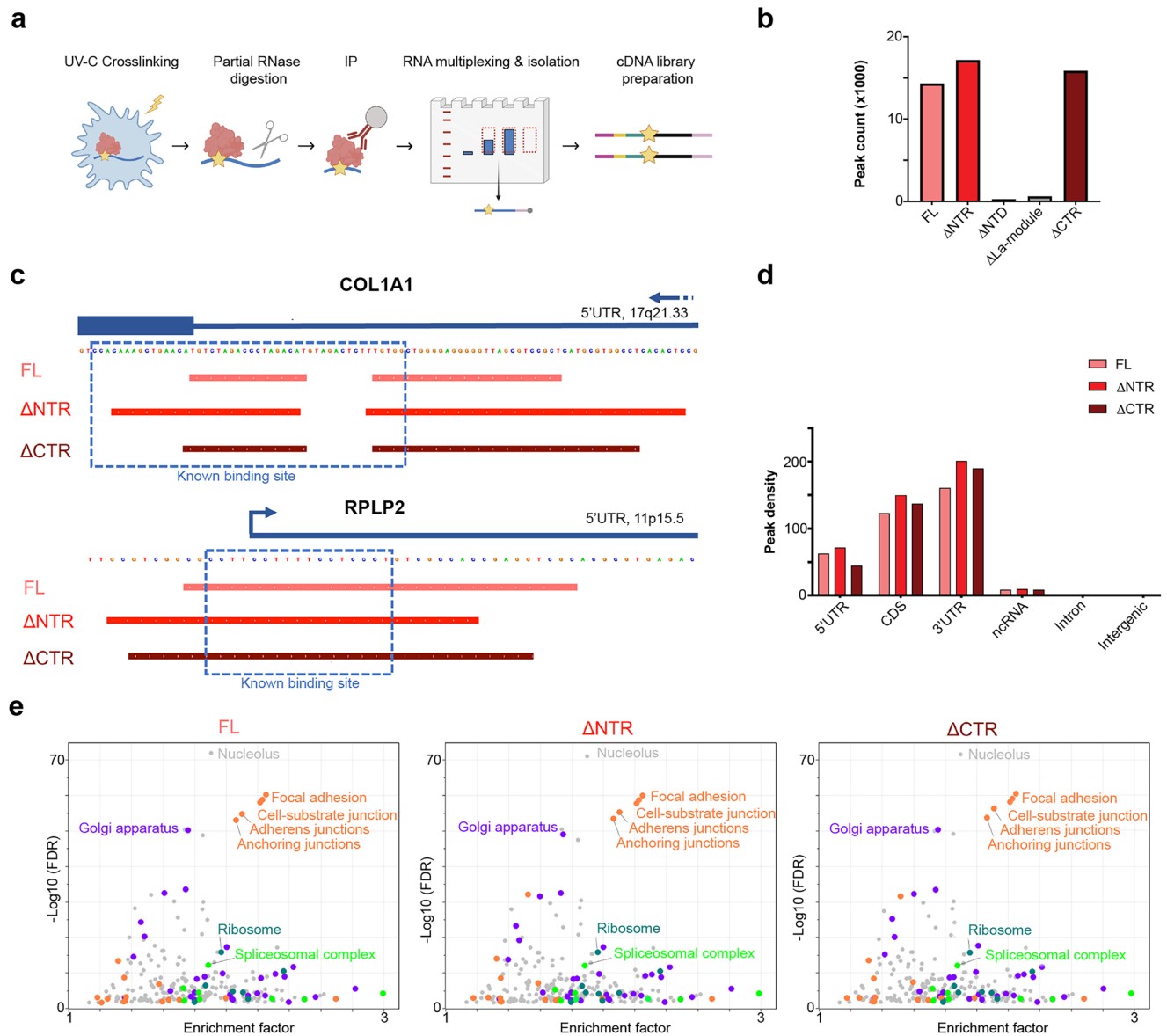

**Fig. 2 | The La-module is essential for the RNA-binding activity of LARP6 in living cells. a** Schematic diagram of the iCLIP workflow (Created with modifications using BioRender. Mardakheh, F. (2026) https://BioRender.com/l6v6db4). **b** Total number of iCLIP peaks (representing distinct binding sites), which were identified for each myc-LARP6 variant. The ΔNTD and ΔLa-module mutants did not bind to RNA in cells and were therefore excluded from further downstream analysis. **c** iCLIP binding profile of indicated myc-LARP6 variants in the 5'UTR of COL1A1 mRNA, as well as the 5'UTR of RPLP2 ribosomal protein mRNA. The identified peak regions with respect to the previously known sites of binding are marked. **d** Normalised distribution of iCLIP peaks across different genomic regions. The peak numbers were normalised to the length of each genomic region to calculate peak density. LARP6 mostly binds to coding transcripts, with the majority of binding occurring in the 3'UTR regions. **e** GO cellular component category enrichment analysis of the FL, ΔNTR, and ΔCTR myc-LARP6-bound transcripts. The iCLIP data was normalised to total RNA-seq data to account for transcript abundance.

CTR are dispensable for the bulk of LARP6 RNA-binding activity in cells.

## The NTR of LARP6 mediates RNA-binding selectivity in living cells

Next, we examined the underlying principles of RNA target recognition by LARP6. For this purpose, we used Pythia, a neural network-based tool designed to model RBP preferences from CLIP data[22]. Pythia utilises context-free grammars (CFG) to capture both sequence as well as structural determinants of RNA binding[23–25]. We used the identified iCLIP peaks as input to train models for FL, as well as ΔNTR and ΔCTR mutants. The data were divided into training (64%), tuning (16%), and validation (20%) subsets with non-overlapping chromosomes among

the 3 data splits. High-quality models were successfully trained for each iCLIP dataset, with the ability to predict each LARP6 variant binding sites with high accuracy and precision (Supplementary Fig. 3a–c). We then evaluated the performance of each model for predicting the binding sites of the other variants, as an indicator of divergence in their RNA binding. Area under the precision-recall curve (auPRC) was calculated as the performance metric for each model. The ΔCTR model exhibited an auPRC value of 86%, when applied to its own validation dataset, but this performance was slightly reduced when the model was applied to the FL and ΔNTR datasets, with auPRC values dropping to 78% and 80%, respectively (Fig. 3a). The ΔNTR model exhibited a similar performance, with an auPRC value of 86% when applied to its own validation dataset, but the performance was slightly more

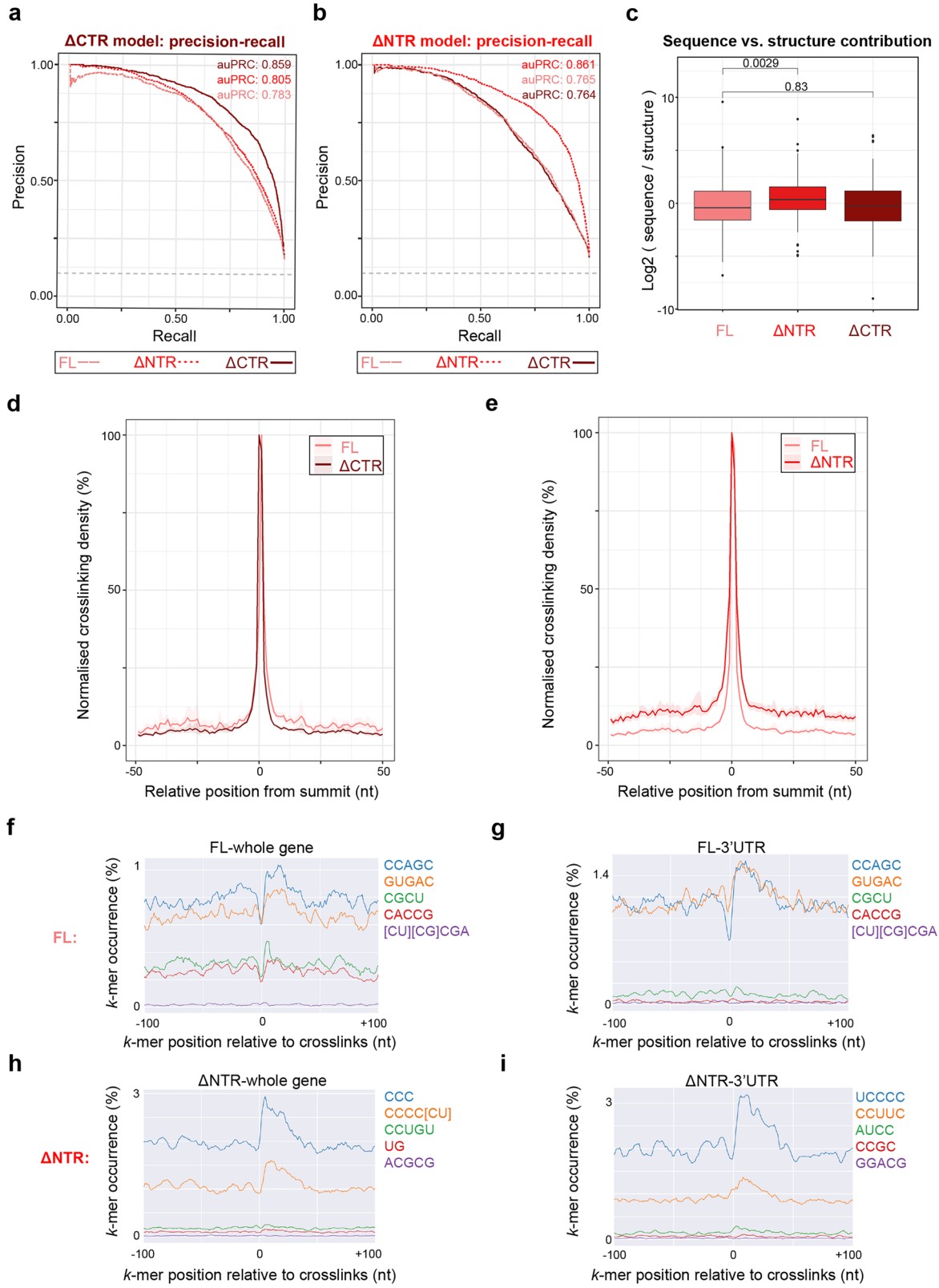

compromised when the model was applied to the FL and ΔCTR datasets, with auPRC values dropping to 76% in both cases (Fig. 3b). These results suggest that both CTR and NTR affect target recognition by LARP6, with the impact of the NTR being slightly more prominent.

Next, we assessed the relative contributions of sequence and structure towards RNA recognition by each LARP6 variant. Consistent

with previous SELEX findings[18], FL LARP6 appeared to leverage on both sequence and structural information for identifying its target sites across the transcriptome, with a slight bias towards structure (Fig. 3c). A similar profile was detected for the ΔCTR mutant, but this balance was significantly disrupted when the NTR was deleted, with the binding activity becoming slightly more sequence-dependent (Fig. 3c). These

**Fig. 3 | The N-terminal IDR of LARP6 fine-tunes its RNA-binding activity in living cells. a** Precision-Recall curves for the ΔCTR Pythia model, applied to the FL, ΔNTR, and ΔCTR iCLIP datasets. Area under the Precision-Recall curve (auPRC) values were calculated as measures of the model's performance in predicting the RNA-binding sites for each dataset. **b** Precision-Recall curves for the ΔNTR Pythia model, applied to the FL, ΔCTR, and ΔNTR iCLIP datasets. AuPRC values were calculated as measures of the model's performance in predicting the RNA-binding sites for each dataset. **c** Comparisons of sequence to structure contributions toward the Pythia model performances for the FL, ΔNTR, and ΔCTR iCLIP datasets. Log2 of DeepLIFT randomised sequence to structure ratio values were plotted (n = 151 (FL), 147 (ΔNTR), and 111 (ΔCTR)). Boxplots mark the lower and upper quartiles. Horizontal lines mark the median. Whiskers extend to the minima and maxima, excluding outliers, which are individually marked. Significance was calculated using a two-sided Wilcoxon rank-sum test. The *p*-values are reported on the plots. **d** Metagene analysis of the iCLIP binding sites from myc-LARP6 FL and ΔCTR. The median of normalised crosslink densities observed in the 50 nt surrounding each peak summit are displayed. The shaded regions represent the 95% confidence interval. **e** Metagene analysis of the iCLIP binding sites from myc-LARP6 FL and ΔNTR. The median of normalised crosslink densities observed in the 50 nt surrounding each peak summit are displayed. The shaded regions represent the 95% confidence interval. **f** Positional motif analysis of FL myc-LARP6 iCLIP contact sites by PEKA[26]. Significantly enriched 5-mer motifs in the 100 nt surrounding the crosslink sites in the whole genome are displayed. **g** Positional motif analysis of FL myc-LARP6 iCLIP contact sites by PEKA[26]. Significantly enriched 5-mer motifs in the 100 nt surrounding the crosslink sites in the 3'UTR regions are displayed. **h** Positional motif analysis of ΔNTR myc-LARP6 iCLIP contact sites by PEKA[26]. Significantly enriched 5-mer motifs in the 100 nt surrounding the crosslink sites in the whole genome are displayed. **i** Positional motif analysis of ΔNTR myc-LARP6 iCLIP contact sites by PEKA[26]. Significantly enriched 5-mer motifs in the 100 nt surrounding the crosslink sites in the 3'UTR regions are displayed.

analyses suggest that the NTR significantly affects how LARP6 recognises its contact sites with RNA, with sequence-based information becoming more prominent in selecting the binding sites.

To gain further insights into the consequences of IDR deletions on LARP6–RNA sites of interactions, we also conducted a metagene analysis of LARP6 crosslinking footprints across its mapped binding sites (i.e. peaks) in the transcriptome. The meta-profile of FL LARP6 crosslinks showed a highly narrow interaction footprint, which remained mostly unchanged upon CTR deletion (Fig. 3d). In contrast, deletion of the NTR led to a clear broadening of the interaction footprints around the main binding site (Fig. 3e), indicating that the NTR is important for enhancing the RNA-binding selectivity of LARP6 in the vicinity of main peaks. Based on these results, we conclude that the observed increase in sequence-dependent RNA-binding bias by Pythia in the ΔNTR mutant is likely reflective of this loss in RNA-binding selectivity, resulting in the emergence of weaker peripheral RNA-protein interactions, which are likely often devoid of any conserved structural RNA features.

Since Pythia analysis revealed a change in the usage of sequence-based information for target recognition by the ΔNTR mutant, we next looked for significant linear sequence motifs that were present in the vicinity of crosslinking sites for FL LARP6 and the ΔNTR mutant. Positionally enriched *k*-mer analysis (PEKA) was employed to identify significantly enriched linear motifs, using the default package settings[26]. We applied the motif analysis to both the full iCLIP dataset, as well as only the 3'UTR regions, where the majority of LARP6 binding sites reside. The FL exhibited a significant enrichment for several relatively complex sequence motifs, an observation in line with the previously published SELEX findings[18] (Fig. 3f & 3g). Loss of the NTR, however, transformed this pattern, with simpler pyrimidine-rich linear motifs dominating the binding profile (Fig. 3h & 3i). A minimal difference was observed between the full dataset and the 3'UTRs, suggesting the observed changes are not restricted to a specific group of binding targets (Fig. 3f–i). Collectively, these results suggest that while the La-module is crucial for mediating RNA–LARP6 interactions, the NTR plays a role in fine-tuning the target selectivity of the La-module.

**In vitro binding analysis validates a role for NTR in RNA-binding selectivity**

To determine whether the impact of NTR on RNA-binding selectivity is an intrinsic property of this protein region, rather than a result of additional cellular factors, we next investigated LARP6 interactions with its RNA targets in vitro. A segment within the 3'UTR of the CTNNA1 mRNA, which encodes for Catenin α1, a cytoskeletal protein that bridges cadherin molecules with actin filaments[27], was selected as the representative target of LARP6 for in vitro analysis. This choice was due to the strong yet highly focused binding of FL LARP6 to this region in the form of two adjacent iCLIP peaks, alongside weaker interactions

detected only with the NTR-deleted mutant (Fig. 4a), a pattern exemplifying our metagene analysis-based observation. We designed 25 nucleotide-long RNA oligonucleotides corresponding to the two primary peaks (P), as well as four additional surrounding peaks located upstream (U) or downstream (D) of the main binding sites (Fig. 4a). As a control, we used a polyC oligonucleotide, which is not expected to bind to LARP6 FL, but could potentially interact with the La-module in the absence of NTR, based on our iCLIP linear motif analysis (Fig. 3f–i). These RNA oligos were biotinylated for use in biolayer interferometry (BLI) analysis[28]. Biotinylated oligonucleotides were subsequently immobilised on streptavidin-coated biosensors to measure the association-and-dissociation kinetics with the purified La-module and NTD, obtained recombinantly from bacteria (Supplementary Fig. 4a & 4b). A control experiment without RNA loaded on the sensors ruled out non-specific protein binding (Supplementary Fig. 4c & 4d). Steady-state analysis was used to generate equilibrium binding curves for each oligo, by plotting the instrument response against protein concentration. The NTD exhibited varying affinities for the different oligonucleotides representing the CTNNA1 peaks, with low- or mid-nanomolar affinity (17.3–61 nM) towards the main peaks, but weaker binding mostly in the high nanomolar range (85–587 nM) to the peripheral upstream and downstream peaks (Fig. 4b & Table 1). In contrast, the La-module showed comparatively similar binding curves, with affinities in the low nanomolar range (0.9–14 nM) across the different CTNNA1 oligonucleotides (Fig. 4c & Table 1). As expected, virtually no binding was detected between the NTD and the polyC oligo, whereas binding to this oligo, albeit with considerably lower affinity, was measured for the La-module (Figs. 4b, 4c, & Table 1). We also assessed the binding of LARP6 NTR region alone to the oligo representing one of the main peaks, which was found to be ~1000 fold weaker than the La-module (Supplementary Fig. 4e, 4f, & Table 1), consistent with our iCLIP findings that in the absence of the La-module, the NTR IDR does not strongly associate with RNA. Microscale thermophoresis (MST) further validated our BLI measurements, confirming low-nanomolar, mid nanomolar, and micromolar affinities of the La-module, NTD, and NTR for P1 RNA, respectively (Supplementary Fig. 4g–l).

Next, we calculated the relative association constants ($K_a$) of the La-module and NTD to each RNA target. In agreement with our iCLIP results, the NTD exhibited a narrow RNA-binding profile, with the tightest affinity for the oligo corresponding to one of the main peaks, and lower affinities for the surrounding oligo peaks, suggestive of selective RNA binding (Fig. 4d). Conversely, the La-module exhibited a wider, less selective profile across the CTNNA1 peak oligos tested, indicative of its reduced binding discrimination in the absence of the NTR (Fig. 4e). Together, these results corroborate our iCLIP findings from living cells, suggesting that the NTR has an intrinsic role in enhancing the selectivity of the La-module.

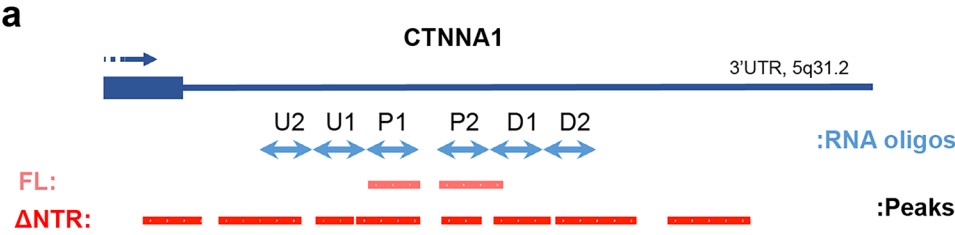

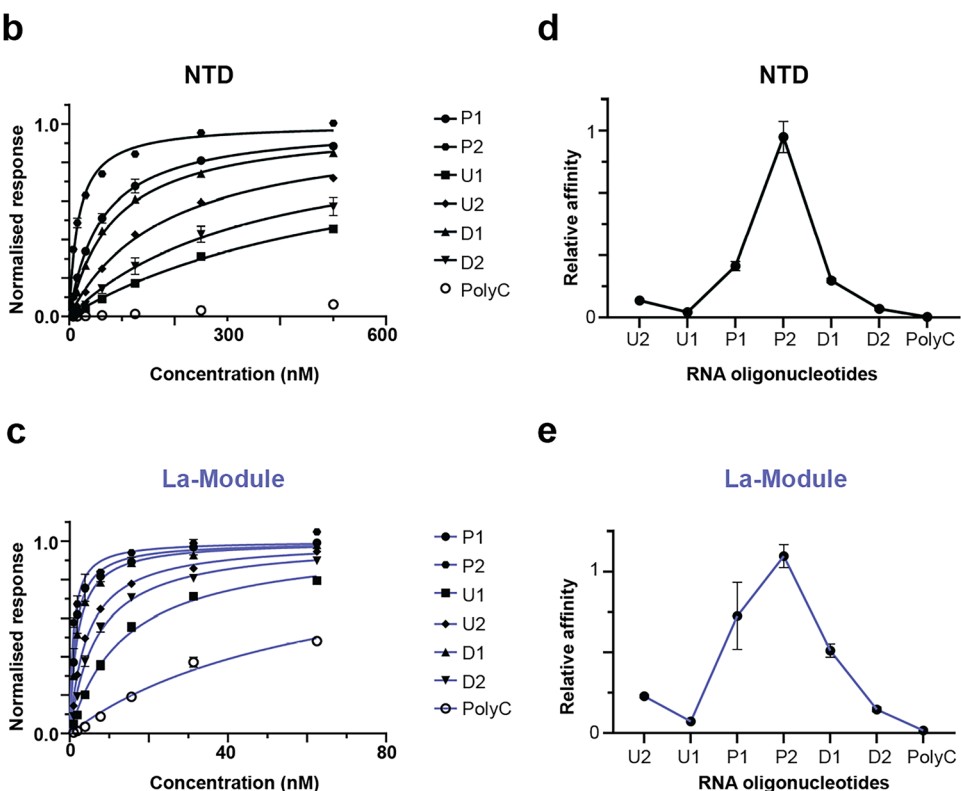

**Fig. 4 | NTR provides binding selectivity to the La-module, in vitro. a** The iCLIP binding profile of FL and ΔNTR myc-LARP6 in the 3'UTR of CTNNA1 mRNA. The 25 nt-long RNA oligonucleotides that are designed to match either the two main FL peaks (P1 and P2) or the 4 peripheral ΔNTR peaks (U1, U2, D1 and D2), are displayed in light blue. **b** Steady-state BLI binding analysis of LARP6 NTD to each indicated 5' biotinylated CTNNA1 3'UTR RNA oligonucleotide. A polyC oligonucleotide was used as control. Error bars = mean ± SD (n = 3 independent biological replicates). **c** Steady-state BLI binding analysis of the LARP6 La-module to each indicated 5'

biotinylated CTNNA1 3'UTR RNA oligonucleotide. A polyC oligonucleotide was used as control. Error bars = mean ± SD (n = 3 independent biological replicates). **d** Relative $K_a$ values, calculated from the BLI steady state analysis of LARP6 NTD (B) against each RNA oligonucleotide. Error bars = mean ± SD (n = 3 independent biological replicates). **e** Relative $K_a$ values, calculated from the BLI steady state analysis of LARP6 La-module (C) against each RNA oligonucleotide. Error bars = mean ± SD (n = 3 independent biological replicates).

**Table 1 | List of BLI-estimated steady-state binding $K_D$ values for the indicated LARP6 protein variants and RNA oligonucleotides**

| RNA oligonucleotide | La-module $K_D$ (nM) | NTD $K_D$ (nM) | NTR $K_D$ (nM) |
|---|---|---|---|
| U2 | 4.4 ± 0.1 | 183 ± 5.0 | |
| U1 | 14.0 ± 0.8 | 587 ± 29 | |
| P1 | 1.5 ± 0.4 | 61.0 ± 5.0 | 1200 ± 500 |
| P2 | 0.91 ± 0.05 | 17.3 ± 1.2 | |
| D1 | 2.0 ± 0.1 | 85.0 ± 4.0 | |
| D2 | 6.9 ± 0.6 | 367 ± 57 | |
| polyC | 63.0 ± 3.7 | >7500 | |

## The dynamic NTR is a gatekeeper for the La-module conformations

To mechanistically understand how the NTR affects the RNA-binding selectivity of LARP6, we next performed a thorough molecular characterisation of the system. To validate the disordered nature of NTR, we first performed far-UV circular dichroism (CD). The CD spectrum of the La-module exhibited the expected profile of a folded protein while the NTR showed a minimum at ~200 nm and very low ellipticity at ~220 nm, indicative of intrinsic disorder[29] (Fig. 5a). In contrast, the spectra of the NTD showed shapes and molar ellipticities compatible with the presence of both folded and unfolded regions (Fig. 5a). For residue-level information on the conformation and dynamics of the NTR, La-module and the NTD, we used NMR spectroscopy. This

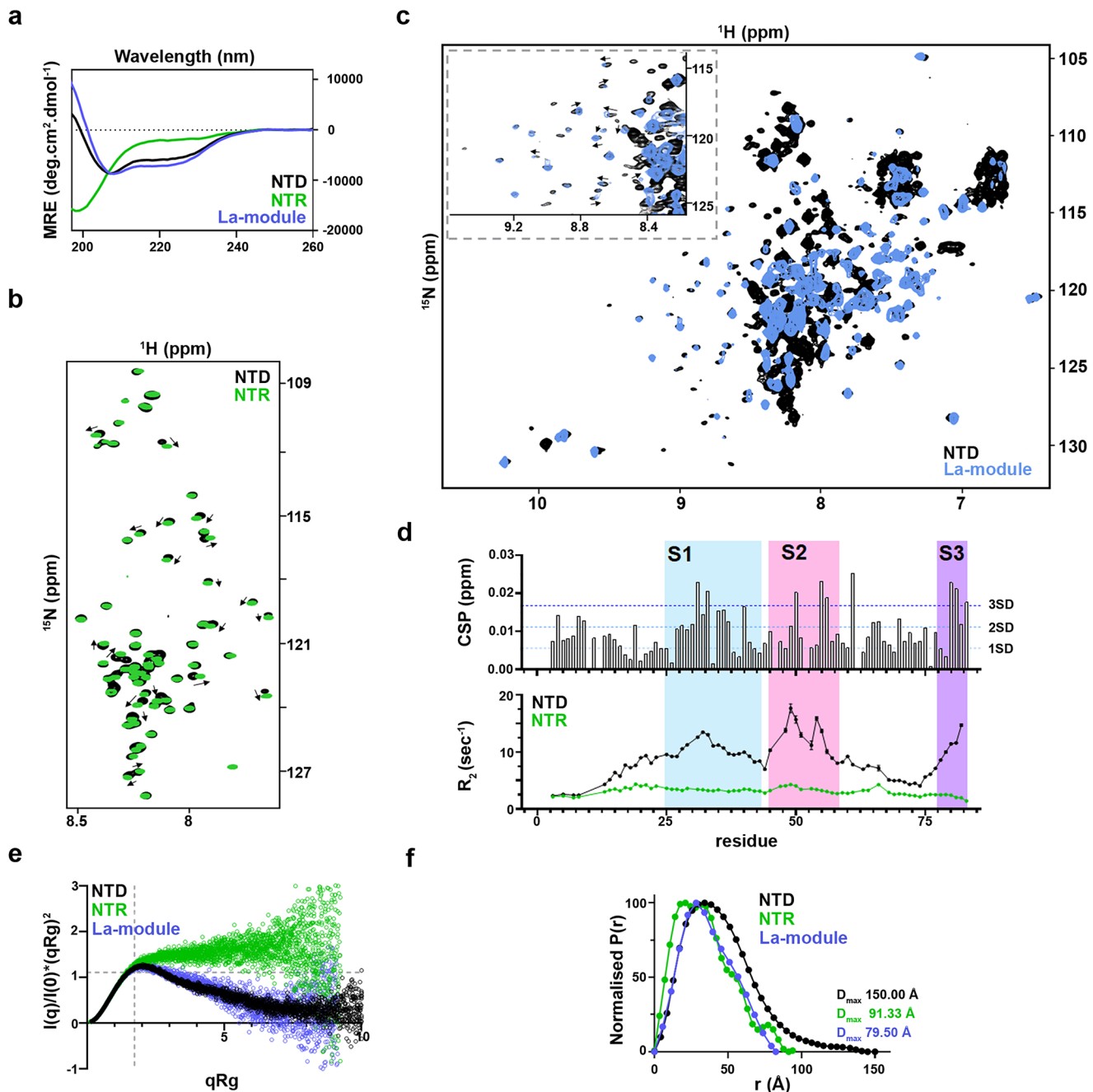

**Fig. 5 | NTR controls the conformational selection of the structured La-module in LARP6. a** Far-UV CD spectra of recombinant LARP6 NTD, La-module, and NTR protein segments. **b** Zoom-in of the overlay of $^1$H-$^{15}$N TROSY NMR spectra of recombinant LARP6 NTD and NTR, in the window of $^1$H-resonances for flexible/unstructured tails. Several peaks undergo CSP (grey arrows) upon NTR tethering to the structured La-module. Full spectra are reported in Supplementary Fig. 5a. **c** Overlay of $^1$H-$^{15}$N TROSY NMR spectra of recombinant LARP6 La-module and NTD. The inset at the top-left corner shows a zoomed-in view in a region where several perturbations of the La-module peaks are observed upon the tethering of the NTR (grey arrows). **d** Analysis of NTR resonances in the context of the NTD. Top: CSP of the residues of the NTR in the context of the NTD. Bottom: $R_2$ NMR backbone relaxation of the residues of the NTR in isolation (green) vs. in the context of the NTD (black). Relaxation rates were determined from exponential least-squares fitting of peak volumes. Error bars indicate mean ± SD, estimated from n = 100 cycles

of Monte Carlo simulation and refitting. Three approximate segments exhibiting larger CSPs and enhanced $R_2$ are highlighted: S1 = residues 25–43; S2 = residues 45–58; S3 = residues 77–83. **e** Overlay of the normalised Kratky plots obtained from the SEC-SAXS data analysis of recombinant LARP6 NTD, La-module, and NTR. The NTR in isolation shows high data points deviation from the typical globular proteins $(I(q)/I(0) \times (qRg)^2 = 1.104$, $qRg = 1.73$–indicated by the grey dotted cross), plateau at high qRg, and large data scattering. The La-module and the NTD exhibit a maximum slightly above the typical value for globular proteins, and slow data points decay, indicative of elongated conformations. The scattering of the NTD datapoints is lower than the La-module, suggestive of specific La-module conformations selection by NTR. Full data sets are reported in Supplementary Fig. 5e–g. **f** Overlay of the P(r) distance distribution functions of recombinant LARP6 NTD, La-module, and NTR. The NTR exhibits multiple states, some of which are highly extended. The La-module shows two main conformations, whilst The NTD exhibits a single state.

allowed us to identify folded domains, corresponding to the La-module, tethered to an unstructured flexible NTR. These features were manifested in the backbone amide chemical shift pattern in the $^1$H-$^{15}$N transverse-relaxation optimised spectroscopy (TROSY) experiments, in that well-dispersed peaks correspond to those assigned in the La-module[30], whereas those arising from the NTR cluster in $^1$H-frequencies between ~7.5 and 8.5 ppm (Fig. 5b & Supplementary Fig. 5a), typical of disordered regions. These were sequentially assigned (Supplementary Fig. 5b). The unstructured and dynamic nature of the NTR resonances was further corroborated by their low secondary structure content and $R_2$ values (Supplementary Fig. 5c–d).

A closer inspection of the $^1$H-$^{15}$N TROSY spectra of NTD versus La-module and NTR in isolation detected numerous chemical shift perturbations (CSPs) both in the resonances corresponding to the La-module and the NTR, reflecting changes of the chemical environment of these amide groups (Fig. 5b & 5c). This indicates direct contacts between the NTR and the La-module residues, as well as any accompanying conformational rearrangement in the tethered NTD fragment (Fig. 5c). Notably, the observed interaction between the two parts exhibits a transient and dynamic character, given the relatively small magnitude of the observed CSPs in the NTR resonances, as well as their retained ensemble features in the context of the NTD (Fig. 5b & 5c). A further examination by NMR backbone relaxation analysis, notably revealed enhanced $R_2$ rates for selected clusters of NTR residues when tethered to the La-module (Fig. 5d), indicative of decreased molecular motion. The overlap of these regions with resonances showing larger CSPs in the NTD context led to the delineation of two segments (S1 and S2), centred around residues 34 and 52 respectively, as directly engaged in La-module contacts (Fig. 5d). Instead, the changes in dynamics and CSPs observed for a third segment (S3) at the end of NTR are likely attributable to its immediate proximity to the La-module junction, although additional points of interaction cannot be excluded.

To dissect further the relationship between the La-module and the NTR moieties, we turned to size exclusion chromatography small-angle X-ray scattering (SEC-SAXS), a sensitive technique for assessing the overall molecular shape and conformational transitions in solution, which is particularly well-adapted for the quantitative analysis of IDRs[31,32]. The NTR, the La-module and the NTD exhibited distinct profiles in their dimensionless Kratky plots[31] (Fig. 5e). The NTR presented a major deviation from the typical bell-shaped curve expected for globular proteins, showing a quasi-monotonic increase to high qRg values (where q is scattering vector and Rg is the radius of gyration), a profile that is typical of proteins with intrinsic disorder[31] (Fig. 5e & Supplementary Fig. 5e). The La-module data recapitulated the behaviour observed previously[33], where the structurally independent domains forming the La-module (La-motif and RRM1) sample the conformational space and exist in discrete subsets of accessible populations in a dynamic equilibrium (Fig. 5e & Supplementary Fig. 5f). Interestingly, although the NTD showed an elongated scattering plot compared to the La-module—attributable to the additional tethered residues of the NTR—it also displayed a more pronounced bell-shaped profile, concomitantly with lower data point scattering, suggestive of an overall reduction in the protein dynamics (Fig. 5e & Supplementary Fig. 5g). These changes can be readily illustrated in the comparative analysis of the pairwise distance distribution function P(r), specifically an increase in the maximum dimension $D_{max}$ in the NTD, and a change in profile (which is a read-out of the macromolecule's shape[34]) to a nearly monomodal distribution (Fig. 5f, Supplementary Fig. 5e–g). Overall, the SAXS profiles indicate a change in the geometry and dynamics of the La-module in the context of the NTD, whilst the NTR retains flexible features as evidenced by NMR experiments. Collectively, these data suggest that the NTR does not fold upon interaction with the La-module; nonetheless its transient interaction locks the La-motif and the RRM1 in a distinct population state, preventing or restricting their ability to sample the conformational space.

## NTR promotes selective RNA binding through a composite molecular mechanism

Conformational selection and dynamics are known to play an important role in RNA substrate recognition for several modular RBPs, including LARPs[35,36]. Equipped with the description of the different conformations of the LARP6 segments in their apo state, we next asked whether the unveiled interplay between the NTR and the La-module may impact LARP6 association with RNA. We focussed on the CTNNA1 P1 RNA probe, which represents a high-score LARP6 iCLIP peak, and binds with high affinity to both the La-module and the NTD (Table 1). Upon establishing a 1:1 RNA–protein binding stoichiometry (Supplementary Fig. 6a), SEC-SAXS methodology was applied to the NTD and the La-module in complex with the RNA. For the La-module, the dimensionless Kratky plot revealed a drastic alteration upon CTNNA1 P1 binding (Fig. 6a & Supplementary Fig. 6b), with the curve becoming bell-shaped and its maximum shifting towards the Guinier–Kratky point ($\sqrt{3}$, 1.104)[34], indicating a non-globular La-module in the apo state that becomes nearly globular in the presence of the P1 RNA. Such rearrangement was corroborated by the P(r) distribution curve of the La-module:P1 complex losing the bimodal character observed in the apo state (Fig. 6b, & Supplementary Fig. 6b), altogether indicating that binding to P1 RNA locks the La-motif and RRM1 in a discrete bound conformation. Notably, the extension of the overall size observed in the protein–RNA sample is compatible with the proposed La-module compaction, considering a possible overhang of the RNA oligo (25 nt long) in the complex (Fig. 6b).

In contrast to the La-module, only minor changes in the Kratky plots and P(r) distance distributions were demonstrated for NTD upon complex formation with the P1 RNA (Figs. 6c, 6d, & Supplementary Fig. 6c). Visual inspection of the data suggested a slightly diminished globularity of the complex, which could be at least in part attributable to the RNA. Collectively, the SAXS derived features of LARP6 in the apo and RNA bound forms support a model in which the La-motif and the RRM1 of the La-module are no longer able to undergo conformational rearrangement upon RNA interaction in the context of the NTD, with the globular discrete configuration populated in the NTD apo state being the one that readily engages with P1 RNA. In other words, the disordered NTR exerts a form of conformational selection onto its adjacent RNA-binding di-domain that tailors it to bind specific target RNAs.

To further dissect at residue-level the role of the NTR in regulation of LARP6 RNA binding, we also applied NMR spectroscopy to the LARP6 proteins in the presence of stoichiometric levels of RNA. $^1$H-$^{15}$N TROSY NMR experiments of the NTD upon addition of CTNNA P1 RNA showed CSPs in the folded portion (La-module), evidence of its substantial engagement with the RNA, as well as in the NTR peaks (Fig. 6e–f & Supplementary Fig. 6d–e). Backbone relaxation analyses of the NTD:P1 RNA complex revealed that the segments S2 and S3 of the NTR do not exhibit large changes in $R_2$ values compared to apo NTD, indicating negligeable alterations in intrinsic motion upon RNA binding for these regions. On the contrary, a marked decrease in $R_2$ values was observed for segment S1 resonances, suggesting that RNA binding to the NTD disrupts the transient association of this part of the NTR with the La-module, resulting in an enhanced dynamic behaviour (Fig. 6g). This interpretation is supported by the negatively charged nature of S1 ($^{25}$AEDVDELEDEEEGAETRGA[37]) that can likely mimic the backbone charge of RNA, and the observation that several S1 resonances fully or partially revert to their chemical shift values held in the isolated NTR (Supplementary Fig. 6f & 6g). Notably, the CSP analysis of the NTD in the presence of RNA further revealed that residues belonging to S2 experience distinct perturbations shifting to new positions (Supplementary Fig. 6f & 6g). Intriguingly, this fragment (residues 45–58) overlaps with one of the IP-OOPS N-terminal regions identified as a potential direct site of crosslinking to RNA in living cells (residues 42–48, Fig. 1c), endorsing the hypothesis of a direct engagement with RNA for this segment of the NTR.

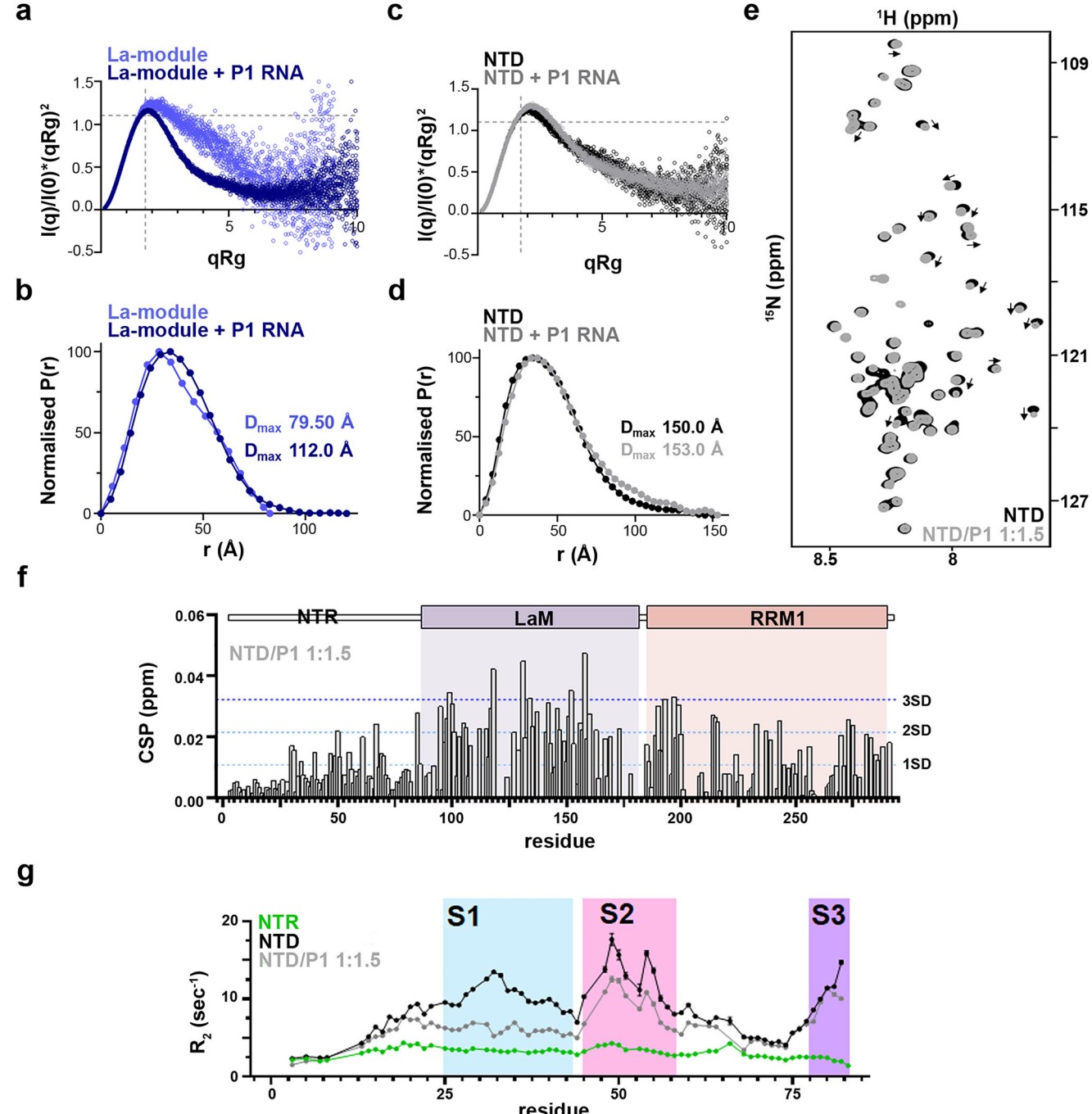

**Fig. 6 | The N-terminal disordered region regulates the RNA-binding properties of LARP6 through a composite mechanism. a** Overlay of the normalised Kratky plots obtained from the SEC-SAXS data analysis of recombinant LARP6 La-module, with or without P1 RNA (1:2 protein:RNA molar ratio). The comparison of plots indicates a conformational rearrangement of the La-module upon P1 RNA binding, leading to a quasi-globular locked conformation of the LaM and the RRM1. Full data sets in presence of RNA are reported in Supplementary Fig. 6b. **b** Overlay of the P(r) distance distribution functions of recombinant LARP6 La-module, with or without P1 RNA (1:2 molar ratio). The overall extension in the P(r) distance distributions is compatible with part of the P1 RNA elongating the shape of the complex. **c** Overlay of the normalised Kratky plots obtained from the SEC-SAXS data analysis of recombinant LARP6 NTD, with or without P1 RNA (1:2 molar ratio). The comparison of plots shows small conformational rearrangements to the NTD upon RNA-binding. Full data sets in the presence of RNA are reported in Supplementary

Fig. 6c. **d** Overlay of the P(r) distance distribution functions of recombinant LARP6 NTD, with or without P1 RNA (1:2 molar ratio). The comparison of plots shows minor conformational rearrangements to the NTD upon RNA-binding. **e** Zoom-in of the overlay of $^1$H-$^{15}$N TROSY NMR spectra obtained after adding P1 RNA into $^{15}$N-labelled LARP6 NTD. The region reports the window of $^1$H-resonances attributable to the flexible/unstructured NTR. Few peaks were perturbed (see grey arrows) upon RNA addition, indicating that the NTR only minorly engages with the RNA. Full spectra are reported in Supplementary Fig. 6e. **f** Per-residue CSP of the NTD resonances upon P1 RNA addition. Residues in both the La-module and the NTR are affected (SD: standard deviation). **g** $R_2$ relaxation rates of the NTR region in isolation, when tethered to the La-module in both apo and CTNNA1 P1-RNA bound states. Relaxation rates were determined from exponential least-squares fitting of peak volumes. Error bars indicate mean ± SD, estimated from n = 100 cycles of Monte Carlo simulation and refitting.

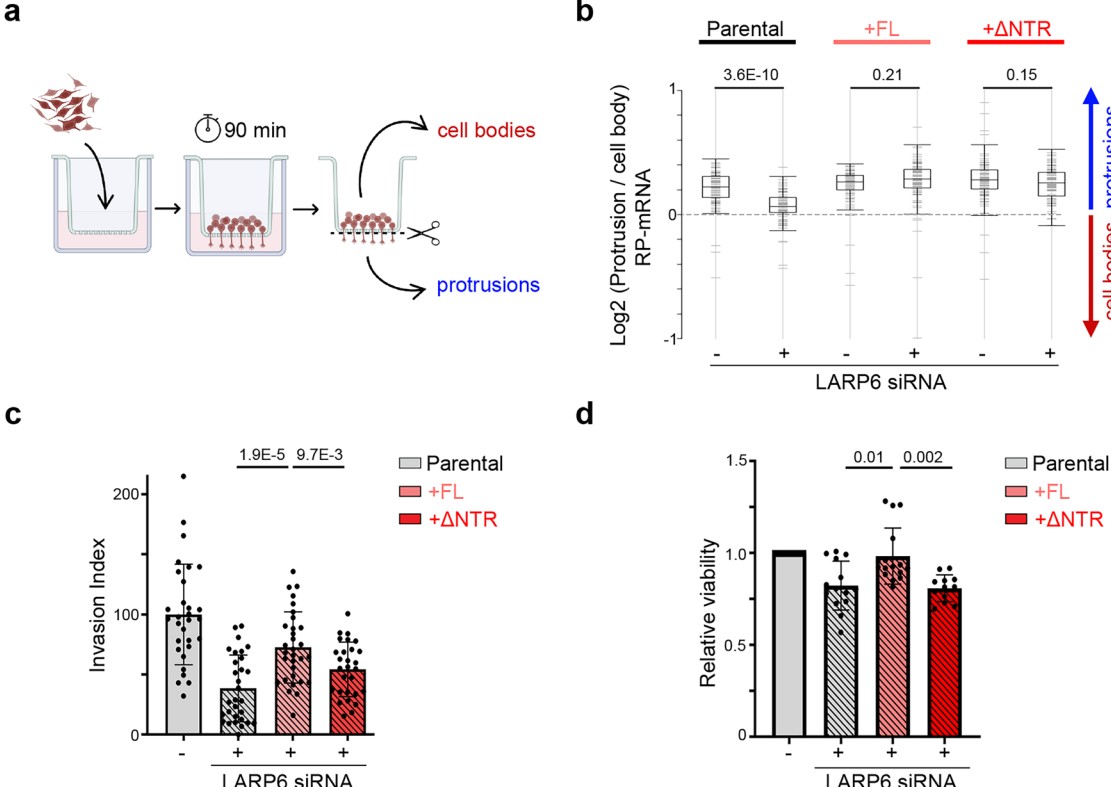

**Fig. 7 | LARP6 NTR is dispensable for RNA localisation, but critical for regulation of cancer cell viability and invasion. a** Experimental workflow for profiling RNA localisation to protrusions and cell bodies of U-87 MG cells, using 3 μm transwell filters. Cells were seeded on top of filters and allowed to form protrusions through the pores for 90 minutes. RNA was isolated from each side of the filter and subjected to 3′ end RNA-sequencing to quantify relative mRNA enrichments in protrusions vs. cell-bodies (Created with modifications, using BioRender. Mardakheh, F. (2026) https://BioRender.com/16v6db4). **b** Profile plot of log2 protrusions/body ratio values for all ribosomal protein encoding mRNAs, in the indicated U-87 MG cells (n = 3 biological replicates). Positive values indicate enrichment in protrusions (blue arrow), while negative values indicate enrichment in cell bodies (red arrow). The boxplots mark the lower and upper quartiles with the median marked by the horizontal line. The whiskers extend to the minima and maxima, excluding outliers. An unpaired two-sided two-sample *t*-test was used for the calculation of significance. *P* values are indicated on the bar graphs. **c** Normalised invasion index

for the indicated U-87 MG cell populations upon endogenous LARP6 knockdown and rescue with doxycycline-inducible myc-LARP6 FL or ΔNTR. The values were normalised to the average non-targeting parental control condition. Error bars = mean ± SD (n = 30 (Parentals), 31 (FL), 28 (ΔNTR) biological replicates, pooled from 3 separate experiments). An unpaired two-sided two-sample *t*-test was used for the calculation of significance. The *p*-values are indicated on the bar graphs. **d** Normalised cell viability for the indicated U-87 MG cell populations, upon endogenous LARP6 knockdown and rescue with doxycycline-inducible FL or ΔNTR myc-LARP6. Cell viability was assessed five days after siRNA transfection, using CTG assay. The values were normalised to the average non-targeting parental control condition. Error bars = mean ± SD (n = 14 (Parental), 16 (FL), 12 (ΔNTR) biological replicates, pooled from 3 separate experiments). A two-sided Kolmogorov–Smirnov test was applied for the significance calculation. The *p*-values are indicated on the bar graphs.

Finally, a comparison of the CSPs observed upon RNA titration revealed that the La-module in isolation undergoes perturbations of greater magnitude and coverage than when it is part of the NTD (Supplementary Fig. 6h). These larger CSPs reflect both direct RNA contacts and the domain–domain rearrangements identified in our SEC-SAXS analysis (Fig. 6a, b), for which the relative contribution cannot be resolved. Together, these results indicate that through locking the La-motif and the RRM1 in a defined conformation, the NTR prepares the La-module for selective engagement with RNA. Nonetheless, RNA binding selectivity imparted by the NTR could also in part derive from direct contacts with RNA via the S2 segment, and through modulating RNA access to the La-module in a competitive manner via the S1 segment.

### NTR-mediated selective RNA-binding is important for LARP6 function in cells

Having revealed the N-terminal IDR of LARP6 as a key mediator of RNA-binding selectivity, we next sought to determine whether this activity was important for LARP6 function in cells. To investigate this, we developed a knockdown-rescue system in which endogenous LARP6

could be selectively depleted using an siRNA oligonucleotide targeting its 3′UTR, followed by rescue with our previously described doxycycline-inducible myc-tagged LARP6 expression system that is siRNA-resistant due to lacking the endogenous LARP6 3′UTR (Supplementary Fig. 7a). Our previous work had demonstrated a role for LARP6 in localising RP-mRNAs to the protrusive fronts of invasive breast cancer cells, as well as supporting cancer cell viability, and promoting invasion[17]. We therefore assessed whether LARP6 played similar roles in U-87 MG cells. Using a microporous filter-based method for isolating protrusions and cell bodies coupled with RNA sequencing[17], we first profiled RNA localisation to protrusions of U-87 MG cells in a global unbiased manner (Fig. 7a & Supplementary Data 4). As expected, RP-mRNAs were enriched in protrusions, but this localisation was significantly abrogated following LARP6 depletion (Fig. 7b & Supplementary Data 4). Notably, rescue of LARP6 expression with either FL or NTR-deleted mutant myc-LARP6 restored the localisation of RP-mRNAs to protrusions (Fig. 5b), demonstrating that this localisation does not significantly depend on the NTR.

Next, we assessed the role of NTR in the regulation of cell migration and invasion, using a 3-dimensional (3D) collagen-I

invasion assay[37] (Supplementary Fig. 7b). Depletion of LARP6 resulted in a strong reduction in the ability of parental U-87 MG cells to invade into the 3D collagen-I matrix (Fig. 7c & Supplementary Fig. 7c). The effect of LARP6 depletion was comparable to inhibition of Rho Kinase, a potent regulator of cell movement[38] (Supplementary Fig. 7d & 7e), highlighting a key role for LARP6 in mediating U-87 MG cell invasion. Crucially, while rescuing LARP6 expression with FL myc-LARP6 could almost fully recover the invasiveness of U-87 MG cells, this rescue was significantly compromised with the NTR-deleted mutant (Fig. 7c & Supplementary Fig. 7c), suggesting that the NTR is required for the full functionality of LARP6 in the context of invasion promotion. We also assessed the effect of LARP6 depletion on the viability of U-87 MG cells. Knockdown of LARP6 resulted in a modest but significant reduction in cell viability, an effect that was rescued by the re-introduction of myc-LARP6 FL (Fig. 7d). The ΔNTR mutant, however, was not capable of rescuing this effect (Fig. 7d), suggesting a key role for the NTR in promoting LARP6-mediated cancer cell viability. Collectively, these depletion-rescue experiments reveal that while the NTR-mediated selectivity in RNA binding may be dispensable for some aspects of LARP6 function, it is likely to be important for the protein's full activity in promoting viability and invasion.

## Discussion

IDRs have emerged as distinct RNA-binding protein modalities in recent years, but their exact contributions towards the RNA-binding activity of most RBPs have remained enigmatic. In this study, we devised an unbiased mass spectrometry-based approach to map the *in cellulo* RNA interacting regions of LARP6, an RBP with a diverse RNA-binding repertoire that is evolutionary conserved in all multicellular organisms[8,17,18]. Our study uncovers a previously unappreciated complexity in the RNA-binding mechanism of LARP6, by demonstrating that in addition to its canonical RBD (the La-module), the protein also engages with RNA *via* its N- and C-terminal IDRs. Our iCLIP analyses revealed that while the La-module remains essential for interacting with RNA in living cells, the N-terminal IDR (NTR) contributes to the RNA-binding selectivity of LARP6. Deletion of this region led to broader and less discriminative RNA interaction footprints, and a shift toward increased sequence-driven binding, consistent with loss of selectivity. These effects were recapitulated in vitro, where the presence of NTR enhanced discrimination among closely spaced RNA-binding targets, underscoring its intrinsic role in fine-tuning the interactions of the La-module with RNA.

Mechanistically, we show that the NTR functions by dynamically locking the La-module into a preferred RNA-selective conformation. Our combined NMR, CD, and SAXS analyses revealed that the disordered NTR interacts with the La-module to constrain its conformational flexibility, whilst itself remaining in a dynamic state. This transient interaction locks the La-motif and the RRM1 of the La-module into a discrete spatial orientation that closely mimics the bound state for a high affinity RNA ligand, facilitating selective filtering of the RNA target sites. To our knowledge, this mechanism of conformational gating of a structured domain by an adjacent IDR has not been reported before. Further to this, the disordered NTR may also contribute to the increased RNA binding selectivity by participating in direct contacts with RNA substrates, as well as modulating access to the La-module. The relative weights of these contributions to RNA binding selectivity remain to be determined. Our findings add a distinct layer of regulatory control to the RNA recognition properties of LARP6, and reveal how IDRs can modulate the specificity and efficiency of tethered RBDs in modular RPBs. Given that LARP6 is known to have a highly diverse RNA-binding repertoire[17,18], the observed conformational flexibility of the La-module likely underpins this versatility. With this regard, NTR-mediated conformational gating could provide a tuneable mechanism for context-dependent recognition of distinct RNA substrates. Notably, the NTR of LARP6 is known to undergo phosphorylation[39]. It is therefore conceivable that this or other similar modifications could be modulating NTR interactions with the La-module, potentially altering substrate selection in response to specific cellular cues, a hypothesis that warrants further investigation.

It is important to note that the role uncovered here for the N-terminal IDR of LARP6 in RNA-binding markedly differs from the N-terminal IDR of its paralogue LARP4A[40]. In both cases, these IDRs do not conform to emerging notions of RNA-binding, encompass sequences that are not low-complexity or rich in positively charged residues, and their RNA-related activity entails crosstalk with the adjacent folded La-module. However, while the NTR of LARP4A contains the most determinants of RNA binding affinity, the NTR of LARP6 only weakly associates with RNA, operating instead by enhancing the RNA-binding selectivity of the La-module through a composite mechanism. The exact mechanism by which these disordered regions execute distinct RNA-binding functions in LARP4A and LARP6 likely resides in their specific sequence-related dynamical ensembles, as well as their physical coupling with the tethered structured domain[41], which requires further investigation.

Crucially, we found that while the NTR was dispensable for LARP6-mediated localisation of RP-mRNAs to cell-protrusions, it was required for the full functionality of the protein in promoting viability and 3D invasiveness of glioblastoma cells. These results suggest that even subtle alterations in the RNA-binding patterns of an RBP, such as broadening of its interaction footprints with the target RNAs, can exert significant effects on post-transcriptional regulation. This notion may provide some explanation for the prominence of germ-line mutations that are associated with human genetic disorders within the IDRs of RBPs[1]. Such mutations are unlikely to completely abrogate the RNA-binding activity of an RBP, which may be developmentally lethal, but instead can impact the subtleties of RNA–RBP interaction dynamics, resulting in emergence of disease-specific phenotypes. As demonstrated here, integration of in-cell approaches (e.g. mutagenesis coupled iCLIP) with machine learning techniques (e.g. Pythia) and biophysical characterisations (e.g. SAXS and NMR) can uncover such nuanced changes in the RNA–protein interaction patterns, and will likely be essential for fully understanding the influence of disease-relevant IDRs in future studies.

Our previous research had identified LARP6 as a key regulator of cancer cell viability and invasion in highly invasive breast cancer cells that have undergone EMT[17]. In this study, we extend these functions to glioblastoma, which are amongst some of the highest LARP6 expressing human cancers. Additionally, LARP6 plays a key role in fibrosis through its binding to and promotion of type I collagen mRNA translation[42]. Given its critical involvement in multiple pathological processes, and its restricted expression pattern in normal human tissues, LARP6 has emerged as a potentially promising therapeutic target[43,44]. Our findings here suggest that pharmacological targeting of the La-module, which constitutes its primary RNA-binding interface of LARP6, would likely provide the most potent therapeutic effect. This conclusion stems from the observation that the deletion of La-module abrogates the RNA-binding activity of LARP6 in living cells, with the much weaker IDR-mediated RNA–protein interactions incapable of compensating for the loss of RNA-binding upon La-module removal. Therefore, rational drug design strategies targeting LARP6 should prioritise the inhibition of the La-module to achieve maximal therapeutic efficacy. However, our NMR and SAXS analyses also reveal that the La-module of LARP6 can exhibit considerable conformational flexibility in a context-dependent manner. As such, this structural plasticity must be carefully considered during any rational drug development efforts, as it may pose challenges to achieving broad, consistent, and effective inhibition.

## Methods

### Cell culture

U-87 MG (ATCC – cat no. HTB-14) and HEK293-T (source: Prof. Chris Marshall's lab, Institute of Cancer Research, London) cells were grown in DMEM media (Gibco) supplemented with 10% foetal bovine serum (FBS) and 1% penicillin/streptomycin. U-87 MG myc-LARP6 inducible cell lines were grown as above, and myc-LARP6 expression was induced with either 60 ng/mL (iCLIP), or 30 ng/mL (all other experiments). MDA-MB231 cells stably expressing GFP-LARP6 cells were generated previously[17], and were cultured as above. All cells were maintained in a humidified incubator at 37 °C, 5% $CO_2$. Cells were passaged consistently before reaching confluence with Trypsin-EDTA (Gibco), and tested regularly for mycoplasma, using the MycoAlert Mycoplasma Detection Kit (Lonza Bioscience).

### Cell line generation

The human codon-optimised LARP6 FL sequence was cloned in the mammalian doxycycline-inducible lentiviral expression vector pCW57.1 using Gateway cloning (Thermo Fisher Scientific) and tagged at the N-terminus with a myc-tag. Deletion mutations were then introduced in the LARP6 FL sequence using Q5 site-directed mutagenesis (NEB) to generate the following deletion mutants: ΔNTR (N-terminal region, deletion of the 2–83 amino acid region), ΔNTD (N-terminal domain, 2–300 deletion), ΔLa-module (La-module, 84–300 deletion), and ΔCTR (C-terminal region, 403–491 deletion). All constructs were validated by Sanger sequencing. Subsequently, each construct was transfected into HEK293-T cells, alongside lentiviral packaging plasmids, using Lipofectamine 2000 (Thermo Fisher), according to the manufacturer's instructions. The lentiviral particle-containing supernatant was collected after 48 h and used to transduce U-87 MG (ATCC) parental cells. The resulting transduced cells were subsequently expanded and underwent selection for ten days with 0.4 μg/mL puromycin, to generate stable doxycycline-inducible myc-LARP6 cell lines. Inducible myc-LARP6 expression was validated and optimised with western blotting and immunofluorescence.

### Endogenous LARP6 knockdown

U-87 MG cells were seeded in tissue culture plates 24 h before transfection. An siRNA selectively targeting the 3'UTR of endogenous LARP6, but not the myc-LARP6 mutants, and a non-targeting siRNA control (scramble) were then transfected into the cells with lipofectamine RNAiMAX (Thermo Fisher), following the manufacturer's protocol, using a final siRNA concentration of 40 nM. Cells were split or harvested 72 h post-transfection for downstream analyses. The knockdown efficiency was validated with western blotting.

### Immunoprecipitation coupled with Orthogonal Organic Phase Separation (IP-OOPS)

Two replicates of 40 million cells were grown in 15 cm dishes and UV-C crosslinked on ice at 400 mJ/cm² in a Hoefer Scientific UV crosslinker. Cells were then harvested by scraping in cold PBS, followed by centrifugation at 500 g for 10 min at 4 °C to obtain cell pellets. Cell pellets were lysed in 500 μL cold RIPA buffer (10 mM Tris-HCl, pH 7.5, 150 mM NaCl, 5 mM EDTA, 0.1% SDS, 1% Triton X-100, 1% sodium deoxycholate, cOmplete protease inhibitor cocktail (Roche), 1 mM PMSF) and incubated on ice for 30 minutes with extensive pipetting on 10 min intervals. Cell lysates were then cleared by centrifugation at 15,000 g for 10 min at 4 °C. GFP-LARP6 was immunoprecipitated using 30 μL of GFP-Trap beads (ChromoTek) for one hour at 4 °C in a rotating wheel, followed by two washes in cold RIPA buffer. Samples were eluted from the beads by direct addition of 1 mL TRIzol (Thermo Fisher) and 5 min incubation at 65 °C, shaking at 1100 rpm. Samples were then placed on a magnet and the eluate was collected. A first OOPS was then performed as described in ref. 19 to isolate the LARP6–RNA adducts. Briefly, phenol-chloroform phase separation was obtained by addition of 200 μL chloroform (SIGMA), extensive mixing, and 15 min centrifugation at 12,000 g, at 4 °C. After centrifugation, three phases had formed: the upper aqueous phase containing free RNA, the lower organic phase containing free protein, and the interphase containing crosslinked LARP6–RNA adducts. The interphase was isolated and washed three times by addition of 1 mL TRIzol, 200 μL chloroform, mixing, and centrifugation as described above. Finally, 1 mL 100% methanol was added to the interphase to precipitate LARP6–RNA adducts, followed by 5-minute centrifugation at 12000 g, at 4 °C, and isolation of the LARP6–RNA pellet. Partial protease digestion with Arg-C (Promega) protease was then performed according to manufacturer's instructions to narrow down the protein regions involved in RNA binding. Briefly, each pellet was resuspended in 150 μL of Arg-C incubation buffer, before addition of the enzyme to the final concentration of 10 ng/ μL. A volume of 16.7 μL of 10X Arg-C activation buffer was then added to each reaction before mixing and incubation overnight (ON) at 37 °C with shaking at 1100 rpm in an Eppendorf ThermoMixer. A second OOPS was subsequently performed, but with some modifications. The samples were first brought to 250 μL by addition of water. 750 μL of TRIzol LS (Thermo Fisher) was then added to the mixture, with the subsequent phase extraction steps carried out as before. Pellets were then resuspended in 50 μL solubilisation buffer (100 mM Tris-HCl, pH 8, 8 M urea) and sonicated with a probe sonicator (SoniPrep 150, MSE, amplitude 10, 15 sec) to help pellet solubilisation. Samples were reduced by treatment with 10 mM DTT (Thermo Fisher) for 30 min at room temperature (RT), followed by alkylation with 55 mM iodoacetamide (G Biosciences) for 30 min at RT, while protected from light. The urea concentration was then diluted four folds by addition of 100 mM Tris-HCl pH 8, followed by the second ON digestion with 2 μg trypsin (SIGMA) in 200 μL of final reaction volume (10 ng/μL Trypsin final concentration) at RT, with shaking at 1100 rpm. The reaction was then stopped by addition of 2% TFA. We also collected the first and second organic phase fractions, precipitated them by the addition of 80% Acetone at −20 °C ON, and subjected the protein pellets to Trypsin digestion as described above, in order to assess the coverage and peptide observability for LARP6 in our experiment.

### LC-MS/MS mass spectrometry sample preparation, data acquisition, and analysis

The IP-OOPS samples were prepared for LC-MS/MS analysis by desalting following the StageTip procedure[45]. Samples were reconstituted in 7 μL buffer A* (2% acetonitrile, 0.1% TFA, 0.5% acetic acid, water), and 6 μL were injected into a Q Exactive Plus Orbitrap mass spectrometer coupled with a nanoflow Ultimate 3000 RSLC nano HPLC platform (Thermo Scientific). Mass spectrometry analysis was carried out as described before[46]. The MaxQuant (version 1.6.3.3)[47] was used for all the peptide searches and quantifications. Raw data files were searched against a FASTA file of the human LARP6 protein sequence from Uniprot (https://rest.uniprot.org/uniprotkb/Q9BRS8.fasta). Enzyme specificity was set to Trypsin. False discovery rates (FDR) were calculated using a reverse database search approach and were set to 1%. Default MaxQuant parameters were used, except for the enabling of the "Match between runs" and "Re-quantify" options. The downstream data analysis was performed with the software Perseus (version 1.6.2.1)[48]. Contaminants were removed by filtering for "Only identified by site", "Reverse", and "Potential contaminants". The remaining identified LARP6 peptides represented the protein regions adjacent to the ones directly involved in RNA binding.

### Transcriptome-wide analysis of RNA localisation

Separation of protrusions and cell bodies was carried out as described before[17,49]. Briefly, 10 cm polycarbonate membrane transwell filters with 3 μm pores (Costar, catalogue number 3420) were coated on both the top and bottom side with collagen I (PureCol, 3 mg/mL) diluted in water 1:30. Coating was performed ON at 4 °C, followed by complete

transwell drying. Endogenous LARP6 was depleted as described above, and 10 million cells per condition were seeded on top of each 10 cm transwell. Media was added underneath the transwells and cells were incubated in the 37 °C humidified incubator for 90 minutes for protrusion induction. The protrusions that formed on the bottom side of the filters were then separated from the cell bodies at the top of the filters and the two fractions were collected in 4% SDS buffer (100 mM Tris-HCl pH 7.5, 4% SDS, RNase-free). Each sample was split into three replicates and total RNA was extracted using TRIzol LS according to the manufacturer's instructions. The total RNA concentration was measured with the Qubit RNA High-sensitivity assay (Invitrogen) on a Qubit 4 Fluorometer (Invitrogen). The quality of the RNA was validated with the Agilent High Sensitivity RNA ScreenTape assay at an Agilent TapeStation 4000 instrument, with all samples having a quality score RIN ≥ 9.1. Library preparation was performed with the Lexogen QuantSeq 3' mRNA-Seq V2 Library Prep Kit FWD with Unique Dual Indices according to the manufacturer's instructions. Each replicate was processed individually to avoid batch effect. 330 ng total RNA in 5 μL volume was used as input. The amplified libraries were quantified and validated with the Qubit dsDNA High-sensitivity and Agilent D1000 high sensitivity ScreenTape assays, pooled in an equimolar ratio, and deep sequenced on an Illumina NextSeq 2000 instrument, using a single-end 100 nt reverse complement workflow. Sequencing was performed at the Barts and The London Genome Centre. The demultiplexed fastq reads were uploaded on the Lexogen Data Analysis Solution, which allowed the automated pre-processing of the data. Adaptors were trimmed and low-quality reads removed using Cutadapt (V1.18) (software developed by Marcel Martin). Trimmed reads were aligned to the human genome (GRCh38, Ensembl release 107) with STAR (V2.6.1a)[50]. Gene counting was performed with FeatureCounts (V1.6.4)[51] and only uniquely mapped reads were considered. The analysis of protrusions/cell bodies enrichment was performed using the Perseus software. The gene counts were first $\log_2$ transformed, and ratios were calculated by dividing the $\log_2$ counts for the protrusions over the $\log_2$ counts for the cell bodies ($\log_2$ ratio protrusions/bodies). The median of each column was then subtracted to normalise the data. Transcripts were then annotated with the KEGG database, and 2D annotation enrichment was performed on the averaged $\log_2$ ratios, as described in ref. 52, to identify categories that were significantly changing.

### 3D collagen I invasion assay

The invasion assay was performed as described[37], with some modifications. μ-Plates 96-well (ibidi) were first coated with 0.2% BSA (diluted in PBS) for at least 30 min at 4 °C. 200 μL of collagen I mix (18 parts PureCol 3 mg/mL (Advanced BioMatrix), 6.66 parts 5X DMEM (Gibco), 1 part 0.1 M sodium hydroxide (SIGMA)) was prepared on ice for each replicate and 20,000 cells were quickly resuspended in the mix and plated. The plates were always kept on ice after cell plating to prevent matrix solidification. At least technical quadruplicates were used in each experiment. After all samples were plated and before the matrix had set, the plates were centrifuged at 300 g for 5 min at 4 °C to gather the cells at the bottom of the wells. The plates were then incubated in the 37 °C humidified cell incubator for three hours to allow the matrix to solidify. 60 μL complete medium was then added to the top of each well, without touching the matrix, to stimulate cell invasion from the bottom of the well towards the top. 1 μM Rho kinase inhibitor 48 (ROCKi)[38] diluted in DMSO was added to the negative control samples. Cells were incubated in the 37 °C incubator for 24 h and allowed to invade into the matrix. After 24 h, cells were fixed by addition of 18.5% paraformaldehyde (diluted in PBS) to each well. Hoechst 33258 (1:250, Invitrogen) was also added simultaneously to stain the nuclei. Plates were incubated for four hours at RT, then stored at 4 °C until imaging, which was performed within one week. Samples were imaged at a Nikon ECLIPSE Ti Spinning Disk confocal microscope

with a plan apochromat VC 20X objective. Hoechst 33258 was excited with a LUN-V 405 nm laser (30% intensity, 300 ms exposure). Six z-stacks were acquired 10 μm apart starting from the bottom of the well where cells had gathered until the furthest distance reached by cells, for a total of 50 μm invasion distance. The following settings were used: 2×2 binning, 540 MHz readout rate, 12-bit and gain 4-bit depth, large image 2×2 fields with 15% overlap (blending). At least two separate images were acquired for each replicate. The nuclei present in each stack were then counted using the Fiji software[53]. First, each image was converted into binary format with the "threshold" command (default settings), then nuclei that were touching were separated with the "watershed" command. The "analyse particles" function was used to count the nuclei in each stack, with size set to 40-infinity and nuclei excluded from the edges. The bottom two stacks were considered the 0 μm distance, as the matrix was not completely flat in some points. Cells with nuclei in the bottom three stacks (≤ 10 μm distance) were considered non-invading, while cells with nuclei in the top three stacks (≥ 20 μm distance) were considered invading. The invasion capability of the cells was calculated as the invasion index, representative of the percentage of cells that were invading. To calculate the invasion index, the number of nuclei in each stack was first summed, then the percentage of cells that were invading and non-invading was calculated. The percentage of invading cells was then converted into an invasion index by setting the parental cell average as the invasion index = 100 and normalising the other samples accordingly.

### Individual-nucleotide resolution UV crosslinking and immunoprecipitation (iCLIP)

iCLIP was performed on U-87 MG myc-LARP6 cells to identify the binding sites of LARP6 at nucleotide resolution, transcriptome-wide. The five myc-LARP6 mutants and the parental negative control were used in the experiment, in triplicates. iCLIP was performed as described before[21,54], with specific myc-LARP6 optimisations. Briefly, 20 million cells per condition were grown in 15 cm dishes and myc-LARP6 mutants were induced with 60 ng/mL doxycycline for 24 h. Cells were crosslinked on ice at 150 mJ/cm² at 254 nm in a Hoefer Scientific UV crosslinker to form covalent bonds between RBPs and RNAs that were in direct contact, then scraped in PBS and pelleted. Cell pellets were lysed in iCLIP lysis buffer (50 mM Tris-HCl, pH 7.4, 100 mM NaCl, 1% Igepal CA-630, 0.1% SDS, 0.5% sodium deoxycholate, complete protease inhibitor cocktail (Roche), 1 mM PMSF), cleared, and diluted to 1 mg/mL in 1 mL volume. Samples were then treated with 0.2 units/mL lysate RNase I (Thermo Scientific, catalogue number EN0602) and 2 μL Turbo DNase (Ambion) for 3 min at 37 °C to partially digest the RNA. 50 μL Dynabeads Protein G (Invitrogen) per sample were first conjugated with 4 μg (10 μL) mouse anti-myc-tag antibody (Cell Signalling, 9B11) for one hour at RT in a rotating wheel, then added to the samples to perform the immunoprecipitation (IP) for one hour at 4 °C in a rotating wheel. After the IP, samples were stringently washed twice with high-salt wash buffer (50 mM Tris-HCl, pH 7.4, 1 M NaCl, 1 mM EDTA, 1% Igepal CA-630, 0.1% SDS, 0.5% sodium deoxycholate), then once with PNK wash buffer (20 mM Tris-HCl, pH 7.4, 10 mM MgCl₂, 0.2% Tween-20). The RNA fragments were dephosphorylated at the 3' end, followed by 3' end ligation of different barcoded adaptors, to allow for downstream multiplexing of each replicate into one sample. The samples were then eluted from the beads, run on an SDS-PAGE gel, followed by isolation of the RNA fragments and subsequent library preparation as described previously. The amplified libraries were quantified and validated with the Qubit dsDNA High-sensitivity and Agilent D1000 high sensitivity ScreenTape assays, pooled in an equimolar ratio, and deep sequenced on an Illumina NextSeq 500 instrument, using a single-end 100 nt workflow. Sequencing was performed at the Barts and The London Genome Centre.

### iCLIP data analysis

iCLIP data was analysed to identify the RNA-binding sites of LARP6 at nucleotide resolution. Raw fastq reads were first demultiplexed based on the experimental barcodes used, to obtain the data from each separate sample. Ultraplex (version 1.1.4)[55] was used for the demultiplexing with default settings. The demultiplexed reads were then uploaded on the webserver iMAPS (imaps.goodwright.com) for primary iCLIP data analysis. The human genome GRCh38.p13 GENCODE V42 was used with the primary assembly comprehensive annotation. Reads were first trimmed with TrimGalore! (software developed by Felix Krueger), to remove remaining adapters and low-quality reads, then pre-mapped to rRNA and tRNA sequences with Bowtie 2[56] and discarded. This step ensured that reads mapping to repetitive regions were not misaligned to mRNA. The remaining unmapped reads were then mapped to the genome with STAR[50]. PCR duplicates were removed with UMI-tools[57] based on the presence of identical UMIs in reads mapping to the same position. The exact crosslink position was then identified as the first nucleotide after the experimental barcode, and this data was downloaded from the web server and processed further offline. The crosslink information was used to call the peaks, which represent the binding sites of LARP6, for each individual replicate. The peak caller Clippy (version 1.5.0)[58] was used with the following settings: rolling mean 15; minimum counts per gene 3; minimum counts per peak 3; prominence 1 (default); peak threshold 0.4 (default). The peaks called for the individual replicates were then combined into one file per mutant by merging the peaks and summing their peak scores using bedtools (version 2.30.0)[59], with the peak scores indicative of the binding strength. Background normalisation was performed by subtracting the peaks that were called in the parental negative control from each combined peak file, using bedtools. The normalised peaks were then annotated to the corresponding transcript and genomic region using a custom script. First, the genomic coordinates for the start and end of the peaks were averaged to obtain the centre of the peak, which was the nucleotide considered for the annotation. This choice was made because peaks overlapping different genomic regions would otherwise be misannotated. The following priorities were then set for the annotation in genomic regions: CDS > 5'UTR > 3'UTR > ncRNA > intron > intergenic > transcript > gene. These priorities ensured that the coding region and untranslated regions were prioritised over more general terms. Peaks in transcripts without a defined name were annotated as "Unknown". Category enrichment analysis was performed using the Perseus software. The normalised peak scores (normalisation described below) were first batch corrected using ComBat[60]. Data was then annotated with the GO database and underwent a Fisher exact test with Benjamini-Hoechberg FDR cutoff of 0.02. The metagene analysis was performed in R (version 4.3.3) using the metagene2 package (version 1.18.0) (software developed by Eric Fournier, Charles Joly Beauparlant, Cedric Lippens, and Arnaud Droit). First, the nucleotide with the highest crosslink score within each peak, here termed "summit", was identified by comparing each myc-LARP6 FL iCLIP peak (peak score > 3) with its corresponding crosslink signal. The myc-LARP6 FL summits were saved in.bed format and used as the regions of interest in metagene2, with a padding of 50 nt on each side (padding_size = 50). Next, the individual crosslink files for each myc-LARP6 replicate were merged and converted to.bam files with bedtools and used as the alignment files in metagene2. Each dataset was normalised to 100% to obtain the normalised crosslink density for each nucleotide surrounding the summit, also accounting for crosslink score. Metagene2 was run with default parameters, with the exception of: strand_specific = TRUE, bin_count = 100, assay = "RNAseq". Positional motif analysis was performed using PEKA (version 0.1.5)[26] to identify LARP6 binding motifs. First, crosslinks from the three replicate files were merged and sorted by position using bedtools. Next, a genome-wide segmentation file that takes into account the different genomic regions was obtained

from the "segment" function in iCount[61]. The human genome GRCh38.p13 GENCODE V42 was used in this analysis. The k-mer length was set to 5 (default).

### Pythia analysis

Pythia is a deep twin-convolutional neural network[22]. One of the arms of the network is a regular trainable convolutional arm capturing sequence context (sequence arm). The other arm is an untrainable fixed dilated convolutional layer capturing base pairing among nucleotides at different distances (structure arm). Pythia integrates the embedding space of the sequence and structure arms to predict RBP binding. We trained Pythia to distinguish peaks that bind LARP6, and its deletion mutant variants, compared to shuffled sequences with a similar dinucleotide distribution. To quantify the structure versus sequence contribution of each peak, we used DeepLIFT[62] to quantify the impact of permuting the input to the sequence and structure arms on the model. We computed the log ratio of the normalised sequencing arm DeepLIFT scores compared to the normalised structure arm DeepLIFT scores. Peaks with higher scores were considered to leverage more from the sequence preference of LARP6, while peaks with lower scores leverage more from the structural preference of LARP6.

### Total RNA-Seq for iCLIP normalisation

Total RNA-Seq was performed on cells grown under the same conditions as the ones used in the iCLIP experiment to normalise the iCLIP results for transcript abundance. The normalisation was performed to ensure that what was observed in iCLIP was due to true differences in binding and not to differences in RNA abundance. Total RNA was extracted with TRIzol, quantified with the Qubit RNA High-sensitivity assay, and the quality was validated with the Agilent High Sensitivity RNA ScreenTape assay, with all samples having a RIN ≥ 9.3. The whole transcriptome RNAseq library preparation and sequencing was outsourced to Novogene. The obtained demultiplexed reads were then trimmed with TrimGalore! (version 0.6.10) to remove the remaining adaptor sequences and low-quality reads. Reads were then mapped to the human genome GRCh38.p13 GENCODE V42 with STAR (version 2.7.10b). Default parameters were used, and gene counts were generated directly from the mapping step by enabling the –quantMode GenerateCounts option. The gene counts from the RNAseq were used for iCLIP data normalisation. The iCLIP peak scores were normalised for transcript abundance according to the following formula: $normalised\ peak\ score = \frac{(peak\ score)}{gene\ count)*1000}$.

### Generation of LARP6 bacterial expression constructs

The human LARP6 sequence was inserted in the bacterial expression vector pET28 using a cloning-by-PCR strategy, and was engineered to contain an N-terminal 6xHis-SUMO-tag in frame with a Tobacco Etch Virus (TEV) protease cleavage site. The LARP6 constructs generated were the following: FL (2–491), NTR (2–83), and NTD (2–300). The deletions were introduced in the LARP6 FL sequence using Q5 site-directed mutagenesis (NEB). Sequences were validated with Sanger sequencing. The La-module (70–300) construct was generated as described in ref. 9.

### Recombinant protein expression and purification

The LARP6 recombinant proteins were produced in *Escherichia coli* Rosetta 2 cells cultured in Luria broth supplemented with antibiotics. Isotopic labelling was obtained by culturing the cells in M9 minimal media enriched with combinations of $^{15}NH_4Cl$ and $^{13}C$-D-glucose, following previously described protocols[63]. Recombinant protein expression was induced with 1 mM isopropyl β-D-thiogalactoside (IPTG), and bacteria were cultured at 18 °C ON in a shaking incubator. Protein purification was performed as described previously[9]. Briefly, bacterial pellets were first lysed in bacterial lysis buffer (50 mM Tris-HCl, pH 8, 300 mM NaCl, 10 mM imidazole, 5% glycerol, 5 μM

lysozyme, complete protease inhibitor cocktail (Roche), 2 mM PMSF), sonicated with a probe sonicator (Fisherbrand) on ice, and cleared by centrifugation at 38,000 $g$ at 4 °C. Samples were initially purified on a Ni-immobilised affinity chromatography (Ni-IMAC) 5 mL His-Trap FF affinity column (Cytiva) previously equilibrated in 50 mM Tris-HCl pH 8, 300 mM NaCl, 10 mM imidazole, 5% glycerol. The proteins were eluted using a linear imidazole gradient from 10 to 500 mM. The tags were subsequently cleaved by ON incubation at 4 °C with a 6xHis-TEV protease, and the untagged proteins were recovered with a reverse nickel affinity chromatography using the Super Ni-NTA affinity resin (Generon). Nucleic acid contaminants were removed by loading the samples on a 5 mL HiTrap Heparin HP affinity column (Cytiva, for LARP6 La-module, and NTD proteins) or a 5 mL HiTrap DEAE FF affinity column (Cytiva, for LARP6 NTR protein), eluting bound species with a linear KCl gradient from 0.1 to 1 M. The final purified products were dialysed in 20 mM Tris-HCl, pH 7.25, 100 mM KCl, 5 mM MgCl$_2$, 1 mM DTT and concentrated to the desired concentration. The protein concentration was measured at 280 nm absorbance and corrected by the theoretical extinction coefficient derived from the ExPASy ProtParam tool[64]. Protein aliquots were snap frozen and stored at −80 °C.

## Circular dichroism (CD)
Recombinant proteins were first diluted in 20 mM Tris-HCl, pH 7.25, 100 mM KCl, 5 mM MgCl$_2$, 1 mM DTT to a concentration of 0.2 mg/mL. UV and CD spectra were acquired on a Chirascan Plus spectrophotometer (Applied Photophysics) under constant nitrogen flush, as described before[40]. Suprasil rectangular cells of 10 mm and 0.5 mm path length (Starna Scientific) were used for the measurements. The experiments were run at 25 °C with 1 nm spectral bandwidth, 1 nm data step-size, and 1.5 seconds time-per-data-point. First, UV spectra were acquired between 230–400 nm to validate the exact concentration of the recombinant proteins. Next, far-UV CD spectra were acquired between 195–260 nm. The UV spectra were corrected for the scattering and smoothened using Savitsky–Golay smoothing with a convolution width of 4 points. The far-UV CD data in millidegrees was then converted to mean residue ellipticity (Θ, deg*cm²*dmol⁻¹), accounting for the exact concentration of each sample calculated from the UV spectra.

## RNA oligo preparation
RNA oligos utilised in this study (Supplementary Table 1) were synthesised by Integrated DNA Technologies (IDT), Horizon Discovery and Merck, incorporating 5' biotin (Bi) and 5' 6-FAM (fluoresceine) modifications for interferometry and microscale thermophoresis experiments, respectively. When necessary, the samples were deprotected according to the sourcing companies' protocols. Lyophilised RNA powders were resuspended in diethyl pyrocarbonate (DEPC)-treated water to create high-concentration stock solutions, which were diluted to working concentrations immediately before the experiments. The concentration of the RNA samples was adjusted using the extinction coefficient values calculated by the sourcing companies.

## Biolayer interferometry (BLI)
BLI experiments were performed on an Octet R8 (Sartorius) instrument, using the Octet BLI Discovery software (12.2.2.20), and Octet BLI Analysis software (12.2.2.4). Recombinant protein concentrations were measured using a V-760 UV-Visible Spectrophotometer (JASCO). The 5' biotinylated RNA oligos and recombinant proteins were diluted in BLI buffer (20 mM Tris-HCl pH 7.5, 100 mM KCl, 0.5 mM MgCl$_2$, 1 mM TCEP, 0.05% Tween-20). The assays were carried out at 25 °C, in 96-well plates (Greiner), with a sample volume of 200 μL and with 1000 rpm shaking. After hydration of the sensors for at least 20 minutes, the biotinylated RNAs were immobilised on streptavidin-coated biosensors (Octet® SA Biosensors, Sartorius) to a final concentration of 0.10–0.15 μg/mL and incubated with varying concentrations of LARP6

NTD, La-module, and NTR. The protein concentrations used ranged between 0–31.3 nM for the La-module protein, and 0–500 nM for the NTD and NTR proteins. Sensor regeneration in between technical repeats was performed by dipping the sensors in RNase-free 1 M NaCl for 30 s, followed by 30 s in BLI buffer, three times. Experiments were performed at least in triplicate, using freshly prepared protein solutions. Control experiments to assess non-specific protein interactions with the sensors were performed by repeating the assays with sensors with no RNA loaded. Equilibrium dissociation constants ($K_D$) for the RNA-protein interactions were estimated by performing steady-state analysis and plotting the instrument response at equilibrium as a function of protein concentration. The data was fitted using the Octet BLI Analysis software assuming a 1:1 interaction and using non-linear least squares regression. Association constant values ($K_a$) were obtained from $K_D$ values using the formula $K_a = \frac{1}{K_D}$.

## Computational disorder predictions
The IUPred2A webserver (iupred2a.elte.hu)[20,65] was used to predict the likelihood of intrinsic disorder in LARP6 based on its amino acid sequence using an energy estimation approach. The IUPred2 algorithm was used to identify disordered protein regions. The prediction type default option of "long disorder" was used, to predict global structural disorder of at least 30 consecutive residues. A context-dependent prediction of disorder was performed with the ANCHOR2 algorithm, which identified disordered binding regions.

## Size exclusion chromatography-small angle X-ray scattering (SEC-SAXS)
SEC-SAXS data were acquired on the B21 beamline at Diamond Light Source[66], as detailed in (Supplementary Table 2). Samples were prepared at concentrations of 5 mg/mL in SEC-SAXS buffer (50 mM TRIS pH 7.25, 100 mM KCl, 2 mM EDTA, 5 mM MgCl$_2$ and 1 mM DTT). For protein–RNA complexes RNA was freshly added achieving a final protein-to-RNA molar ratio of 1:2. The samples were loaded onto a KW 402.5 FPLC column (Shodex) equilibrated with SEC-SAXS buffer. These samples were then exposed to an X-ray beam with a wavelength of 0.9464 Å and energy of 13.1 keV, produced by a bending magnet source with a flux of 4×10¹² photons/sec and a Q range of 0.0045–0.34 Å⁻¹. The beam centres (X and Y) were located at 129.53 mm and 19.92 mm, respectively, with a size of 0.05×0.05 mm (FWHM) at the detector and 1×0.25 mm at the sample site. Data collection was performed using an EigerX 4 M detector (Dectris) placed 3702.5 mm from the sample, with the experiment conducted at 15 °C. A total of 600 frames were collected at a rate of one frame per second. Data reduction and analysis were carried out using ScÅtter IV[67], following previously reported guidelines[68]. Briefly, frames corresponding to the species of interest were merged after buffer subtraction, including frames with similar radius-of-gyration (Rg) values. Scattering data were processed to minimise residual variation, ensuring Guinier fits did not exceed the qRg maximum limit of 1.3. Flexibility analysis was conducted using Porod–Debye, Kratky–Debye, and SYBILS plots, following established guidelines[68]. Dimensionless Kratky plots provided semi-quantitative insights into the macromolecule's structural state, normalised for mass and concentration. Real-space representations were derived from SAXS curves in reciprocal space. The particle's maximum dimension (D$_{max}$) was modelled using Moore analysis to generate the P(r) distribution, which describes all intraparticle distances, including D$_{max}$. The P(r) distributions were normalised to their respective integrated areas. The final P(r) distribution was accepted only when outliers constituted less than 5% of the data points.

## Nuclear Magnetic Resonance (NMR)
NMR studies were performed using Bruker AVANCE IIIHD and AVANCE NEO spectrometers operating at 16.4 T, 18.8 T and 22.3 T. Transverse relaxation optimised spectroscopy (¹H-¹⁵N TROSY)[69] experiments were

conducted at 293.15 K using $^{15}$N-labelled samples of LARP6 La-module, NTR, and NTD, at a final concentration of 100 μM, in a buffer containing 25 mM TRIS pH 7.25, 200 mM KCl, 0.2 mM EDTA, 5 mM MgCl2, 1 mM TCEP, 0.1 % Tween-20, and 5% D2O (NMR buffer). NMR spectra were processed using TopSpin® (Bruker) and NMRPipe[70], and analysed with CcpNmr V3[71] and NMRPipe[70]. $T_2$ relaxation experiments were recorded using the approach of Yuwen and Skrynnikov, which permits more accurate measurement on IDRs[72], and analysed with CcpNmr V3[71], by exponential least-squares fitting of peak volumes over the course of the data collection. Where overlap in the signals prevented accurate measurements of peak volume, residues were excluded. Average rates and standard deviations were estimated from 100 cycles of Monte Carlo simulation and refitting.

## NMR Backbone Assignment

LARP6 NTR backbone resonance assignment was achieved using $^{13}$C-$^{15}$N-labelled samples at a concentration of 400 μM in NMR buffer, through a combination of 2D $^{1}$H-$^{15}$N HSQC[73], and 3D HNCACB, CBCA(CO)NH, HNCO, HN(CA)CO, HBHA(CO)NH[73,74]. All 3D spectra were acquired at 22.3 T with 20%-40% non-uniform sampling (NUS) of the indirect dimensions. Spectra were processed with NMRPipe[70] using SMILE[75] for NUS reconstruction and visualised using TopSpin® (Bruker) and CcpNmr Analysis Version 3[71]. Approximately 80% of amide groups were assigned confidently in a first round of automated (MARS) assignment[76]. Minor difficulties resulted from ambiguity in residues of two low-complexity regions of the NTR $^{29}$DELEDEEE$^{36}$ and $^{59}$EEEP$^{62}$, as well as attenuated NH signal intensities observed towards the N-terminus. These regions were resolved with two HN(C)N experiments[77], with $^{1}$H and $^{15}$N frequency labelling for the indirectly detected amide acquired in separate spectra. A second round of automatic backbone assignment with MARS, with some additional manual curation, enabled essentially complete assignment of backbone resonances, excluding the N-terminal methionine and the amide group of Ala-2, which was absent in all amide-detected spectra, presumably due to rapid exchange with the solvent. The 5 proline residues were identified in the 2D CON heteronuclear correlation[78] due to their separated $^{15}$N frequency and assigned via the $^{13}$C' frequency of residue (i-1). Secondary structure propensities (SSP) were predicted with the SSP package (https://pound.med.utoronto.ca/JFKlab/index.php#SSP)[76], using the CO, Cα, Cβ, Hα chemical shift deviation from expected random coil values.

## Chemical shift perturbation (CSP) analyses

$^{1}$H and $^{15}$N resonances of the La-module and NTD of LARP6 were assigned with reference to those of the NTR (this work), La-motif and RRM1 (BMRB entries 25159 and 25160[30]). CSPs were calculated using the following equation:

$$CSP = \sqrt{(\delta_{1H})^2 + (0.15 \cdot \delta_{15N})^2} \qquad (1)$$

where δ1$_H$ and δ15$_N$ are the chemical shift differences in the $^{1}$H and $^{15}$N dimensions, respectively.

## Protein–RNA interaction studies by NMR

For RNA interaction studies, protein–RNA complexes were prepared by mixing $^{15}$N-labelled LARP6 La-module and NTD and $^{13}$C-$^{15}$N-labelled NTR samples (100 μM) with CTNNA1 P1 RNA in NMR buffer, to a protein-to-RNA molar ratio of 1:1.5. $^{1}$H-$^{15}$N TROSY experiments were recorded. NMR spectra were processed using TopSpin® (Bruker) and NMRPipe[70], and analysed with CcpNmr V3[71] and NMRPipe[70]. CSPs were calculated as for above.

## Microscale thermophoresis (MST)

Microscale thermophoresis (MST) experiments were conducted using a Monolith NT.115 instrument (NanoTemper Technologies). Binding reactions were prepared in a buffer composed of 20 mM Tris pH 7.25, 100 mM KCl, 1 mM DTT, and 0.05% Tween-20, with a total reaction volume of 20 μL. Concentration ranges of 0.063–32 nM, 15–960 nM and 196–50,000 nM were tested for LARP6 La-module, NTD and NTR, respectively. After preparation, each protein solution was mixed with 5′ 6-FAM-labelled CTNNA1 P1 (Merck) at a final concentration of 25 nM. Samples were loaded into standard treated capillaries (NanoTemper) and MST experiments performed at 25 °C, using 50% LED power and 20% MST power. All measurements were performed in four experimental replicates. Thermophoresis was recorded starting with a 5-second baseline measurement, followed by 30 s of irradiation, and a 5 s recovery period. Normalised fluorescence intensity ($F_{NORM}$) and normalised dose responses were obtained with the MO.Affinity Analysis v2.2.4 software (NanoTemper). Dissociation constants ($K_D$) were determined from fitting the normalised dose responses at different protein concentrations, with a nonlinear binding curve (modified Hill equation) with h = 1.5 and $B_{max}$ < 1.5, using GraphPad Prism.

## Size-exclusion chromatography with multi-angle light scattering (SEC-MALS)

The molar mass of LARP6 La-module and NTD in their apo and bound states with CTNNA1 P1 RNA was estimated by SEC-MALS experiments, using an Agilent HPLC 1260 Infinity II system in line with a DAWN MALS detector (Wyatt Technology™) and a differential refractive index (dRI) Optilab detector (Wyatt Technology™). Separations were carried out using a WTC-015S5 column (Wyatt Technology™) in isocratic mode in SEC-MALS buffer (50 mM Tris pH 7.25, 100 mM KCl, 5 mM MgCl$_2$, and 1 mM DTT), at a flow rate of 0.8 mL/min. Bovine serum albumin (Thermo Fisher) was used to calibrate the system. A total of 100 μL of protein samples, prepared in SEC-MALS buffer at a concentration of 2 mg/mL, was injected, and separation was performed for 25 min. Protein–RNA complexes were prepared at a protein concentration of 2 mg/mL and a 100 μL final volume in SEC-MALS buffer, with a protein-to-RNA molar ratio of 1:1. All experiments were conducted in triplicate. Experimental data were recorded and processed using ASTRA One-Click MW™ using the theoretical extinction coefficient derived from the ExPASy ProtParam tool[64].

## Western blotting

Western blots were performed as described previously[21]. Briefly, cell pellets were lysed in SDS buffer (100 mM Tris-HCl pH 7.5, 4% SDS), sonicated with a probe sonicator (SoniPrep 150, MSE, amplitude 8, 10 sec), and cleared. Total protein concentration was quantified using the Pierce BCA Protein Assay Kit (Thermo Fisher). 10–15 μg total protein was prepared for western blotting by addition of 1X NuPAGE LDS Sample Buffer (Thermo Fisher) and 100 mM DTT (Thermo Fisher). Samples were denatured by boiling at 95 °C for 5 minutes. Samples were then loaded on 4–12% NuPAGE Bis-Tris Midi protein gels (Thermo Fisher) and run in NuPAGE MOPS running buffer (Thermo Fisher) at 120 V for at least 90 minutes. Transfer was performed using Immobilon-P PVDF membranes (Millipore) in transfer buffer (1X Tris-glycine, 20% SDS) for two hours at 1 A, at 4 °C. Membranes were then blocked in 5% milk (diluted in 0.05% PBS-Tween-20) for one hour at RT. Incubation with the primary antibody diluted in 5% milk was performed on at 4 °C, followed by three 10 min washes of the membranes with 0.05% PBS-Tween-20. HRP-conjugated secondary antibodies were then diluted in 5% milk and incubated for one hour at RT, followed by three 10 min washes with 0.05% PBS-Tween-20. Membranes were probed with SuperSignal West Pico PLUS chemiluminescence substrate (Thermo Fisher) and imaged on an Amersham Imager 600 (GE Healthcare). Band intensities were detected using Fiji.

## Immunofluorescence

Western blots were performed as described previously[17]. Briefly, 46000 cells were plated in glass coverslips placed in 12-well tissue

culture plates and allowed to grow for 24 hours. Cells were then fixed in 4% paraformaldehyde (diluted in PBS) for 10 minutes and permeabilised in 0.2% Triton X-100 (diluted in PBS) for 15 minutes. Following three washes with PBS, cells were blocked in 4% BSA (diluted in PBS) for 30 minutes at RT. Primary antibodies were diluted in 4% BSA and incubated for one hour at RT, followed by three washes with PBS. Fluorophore-conjugated secondary antibodies and Hoechst 33258 (1:2000, Invitrogen) were diluted in 4% BSA and incubated for one hour at RT, followed by three washes with PBS. Coverslips were then washed in water and mounted on slides using ProLong Diamond Antifade Mountant (Invitrogen). Coverslips were imaged at a Nikon ECLIPSE Ti Spinning Disk confocal microscope with a plan apochromat λ 60X/1.40 oil objective. Fluorophores were excited with LUN-V 405 nm, 488 nm, and 640 nm lasers. Three z-stacks were acquired 10 μm apart. The following settings were used: 2×2 binning, 540 MHz readout rate, 12-bit and gain 4-bit depth. Five or more images were acquired for each replicate and analysed using Fiji.

### Cell Viability assay

Cell viability was measured with the CTG 2.0 Luminescent Cell Viability assay (Promega). 5000 cells were seeded in 96-well plates at least in quadruplicates in 100 μL volume and allowed to grow for 72 h. The CTG reagent was diluted 1:2 in PBS and 100 μL of diluted CTG reagent were added to each well and mixed. The plate was then incubated on a shaker for 2 min, followed by 10 more minutes at RT, protected from light. Samples were then transferred from the tissue culture-grade 96-well plate to a white CELLSTAR 96-well plate (Greiner) suitable for luminescence measurements. Luminescence was measured at a FLUOstar Omega Microplate Reader (BMG Labtech). The background luminescence signal from a well containing media only was subtracted from each sample luminescence signal.

### Statistical analysis

The statistical test applied to each experiment is indicated in the corresponding figure legend. Analysis of RNA localisation, invasion, colony formation, and CTG assays was performed in GraphPad Prism version 10. A $p$-value cutoff of 0.05 was used. Normality of the datasets were assessed using Prism's normality test. If data distributions were determined to be normal, unpaired t-test was used for significance assessments. Otherwise, a Kolmogorov–Smirnov test was applied for statistical testing of non-normally distributed datasets.

### Data analysis and display

Unless otherwise stated, all quantitative data were analysed and displayed using PyMOL Molecular Graphics System 2.0 (Schrödinger, LLC), Microsoft Excel, GraphPad Prism (Dotmatics), and Biorender.com. Individual data-points in each bar graph represent independent biological replicates.

### Reporting summary

Further information on research design is available in the Nature Portfolio Reporting Summary linked to this article.

## Data availability

Mass spectrometry raw files and their associated MaxQuant output files were deposited on ProteomeXchange Consortium[79] via the PRIDE partner repository, under the accession number PXD064029. RNA-sequencing FASTQ raw files were deposited to the NCBI Gene Expression Omnibus (GEO), under the project accession numbers GSE297587 and GSE316154. The NMR assignment of the NTR were submitted to the Biological Magnetic Resonance Bank (BMRB), under the entry number 53427. The SAXS data were submitted to the SasBDB[80], under the entry numbers SASDYW5 (La-module), SASDYZ5 (La-module/P1), SASDYX5 (NTD), SASDY26 (NTD/P1) and SASDYY5 (NTR). PDB codes of previously published structures used in this study

are 2MTF and 2MTG. Reference BMRB entries used in this work are 25159 and 25160. Source Data are provided as a Source Data file. Source data are provided with this paper.

## Code availability

All code for the Pythia software package can be accessed via https://github.com/goodarzilab/pythia. The custom script for annotation of normalised iCLIP peaks to their corresponding transcript and genomic regions can be obtained from https://github.com/alejrb/iclip-annotation/.

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

## Acknowledgements

We would like to thank the past and present members of the Mardakheh and Conte labs for their useful comments and suggestions on various aspects of the project. We acknowledge the BCI mass spectrometry core facility for their support with the MS experiments, and the Barts Genome Centre for next-generation sequencing. We also acknowledge the Diamond Light Source for time on Beamline B21 under Proposal MX32787, and thank the beamline staff for their support, especially Mr Nikul Khunti. We thank the Centre for Biomolecular Spectroscopy at King's for NMR, CD and MST facilities, funded by the Wellcome Trust and British Heart Foundation awards to M.R.C. and others (ref: 202767/Z/16/Z and IG/16/ 2/32273). The NMR work was further supported by the Francis Crick Institute through provision of access to the MRC Biomedical NMR Centre. The Francis Crick Institute receives its core funding from Cancer Research UK [CC1078]; UK Medical Research Council [CC1078]; Wellcome Trust [CC1078]. This project was supported by a Medical Research Council (MRC) project grant (ref: MR/W001500/1), and a Biotechnology and Biological Sciences Research Council (BBSRC) project grant (ref: BB/X007820/1) to F.K.M, and a Leverhulme Trust grant (ref: RPG-2020264) and a BBSRC project grant (ref: UKRI1921) to M.R.C. A BBSRC London Interdisciplinary Biosciences Consortium (LIDo DTP) PhD studentship (ref: BB/M009513/1) supported F.C.

## Author contributions

F.K.M. and M.R.C. conceived the study, acquired the funding, and supervised the work. G.A. and A.I. performed the SEC-MALS and SAXS experiments. G.A., P.J.S., and G.K. performed the NMR experiments. G.A., A.I., P.J.S., and G.K. analysed the NMR results. P.J.S. performed the backbone NMR assignment of NTR. F.C. performed and analysed the BLI experiments, assisted by G.A. and L.M. G.A. performed and analysed the MST experiments. M.K. and H.G. developed Pythia and performed the LARP6 Pythia analysis. T.T.T.B. assisted with the CD analysis. A.B. assisted with the iCLIP data analysis. F.C. performed all other experiments and data analyses. F.K.M., M.R.C., F.C., and G.A. wrote the manuscript.

## Competing interests

The authors declare no Competing interests.
