## [Transparent Peer Review file · Nature Communications]

An intrinsically disordered region mediates RNA-binding selectivity and cellular activities of LARP6.

Corresponding Author: Professor Faraz Mardakheh

Version 0:

Reviewer comments:

Reviewer #1

(Remarks to the Author)

The manuscript “an intrinsically disordered region mediates RNA-binding selectivity and pro-oncogenic activities of LARP6” by Capraro et al. describes a potential RNA-binding mechanism of LARP6 and how this mechanism influences its function in cancer cells. Using iCLIP and a mass-spectrometry based method to derive RNA specificity and identify the RNA binding region of LARP6, respectively, the authors speculate about the involvement of the IDR at the N-terminal region (NTR) in RNA specificity, as the data shows that deletion of the NTR decreases RNA specificity. They could demonstrate that this is true using BLI to derive affinities between protein constructs and a specific RNA 25-mer derived from one of the iCLIP hits. They already speculate here that the La-motif and the RRM (La-module) sample a large conformational space without the NTR and could therefore bind to more RNA sequences as compared to the full-length protein in which the NTR is thought to arrest the La-module in a certain conformation, which is specific for a certain RNA sequence. This they tried to validate with SAXS and NMR. Later they also attempted to show that a deletion of the NTR influences invasion and viability of cancer cells.

Disclaimer: the iCLIP and mass spectrometry data look sound and are partly well-validated with the in vitro data. However, I have not much hands-on experience on these experiments and the necessary statistics needed and can therefore not comment on this part of the manuscript.

While this story is well-written and appealing (and highly interesting, as there is still little known about RNA specificity beyond single classical RNA binding domains), the evidence for the presented mechanism is not convincing enough and it does not rule out other potential explanations (mechanisms) for the observations made. Before this manuscript can be considered at Nature Communications I recommend looking into the following major concerns I have:

1. The mechanism, the authors presented, by which the NTR influences LARP6's RNA binding specificity is hard to imagine and not fully backed up by the data. Or in other words, other explanations could not be ruled out sufficiently. The authors claim that the La-module (La-motif + RRM) achieves higher RNA sequence specificity through conformational dynamics restriction imposed by the NTR. While I agree that the data shows that the NTR is needed for RNA specificity (BLI data and iCLIP, mass-spec), I am not convinced that it reduces the conformational sampling of the La-module and thereby increases RNA sequence selectivity. For single-stranded RNA, being very dynamic (maybe not in a full-length 3'UTR scenario but surely within a 25-mer), why should a rigid La-module be more sequence-specific than a mobile La-module (imagining that the La-motif tumbles independently of the RRM1)? If it would be about binding ssRNA within a certain RNA structure, I could imagine this to be a mechanism (e.g. Hollmann et al. 2020, Cell Reports), but with a 25-mer in vitro I doubt that. Other possibilities: i) the NTR is involved in RNA binding directly and increases specificity, ii) the NTR partly obscures the RNA binding region of the La-module and thus allows only certain RNA sequences to stay long enough to outcompete transient interactions of the NTR with the La-module (e.g. Kang HS et al., 2020, PNAS), the authors claim that this interaction is transient, which it seems to be indeed if there is an interaction (see minor point below). The evidence from NMR and SAXS is not conclusive. Upon RNA binding, the La-module and the NTD (NTR + La-module) are compacted as evident from SAXS, so I would agree that RNA rigidifies the independent tumbling of the La-motif and the RRM domain, as shown also for many tandem RRM domain proteins by SAXS and NMR (plenty of references). The NTR has a similar influence on the Kratky plot on the La-module without RNA present but not to such an extent. Comparing SAXS (Kratky and $P(r)$) of the La-module, NTR and NTD, it could be that the NTR binds transiently to the La-module, which would lead to a similar observation. Upon RNA binding, the NTR within the NTT could also bind to the RNA and thus increases compaction. The

NMR data is also not conclusive. While the NTR is an IDR (the HetNOE data should be presented as a bar plot to show values between -1 and 1, HetNOE divided by reference, even without assignment, although assignment of the NTR should be straightforward, see below), the chemical shift perturbations (CSPs) between the NTR without La-module and with La-module are indeed very weak. The few CSPs could be in the connecting regions between NTR and La-module, which of course would shift when together in one construct compared to the separate ones (however, these shifts are likely larger). So why did the author not perform the backbone assignment of the NTR. This should be easily possible with a 250 μ M sample and this high-quality HSQC spectrum. Then the HetNOE data could be fully analyzed and the interaction with the La-module could be resolved at residue resolution (see below, mutational analysis). Similar for the La-module. The assignment is already available (as shown in the RNA titration experiment), why not identifying the interacting residues on the La-module side? Then the authors could prepare point mutations on the NTR and La-module side to show that upon mutation the interaction between NTR and La-module is gone, and the RNA specificity should be also affected. Then the mechanism could be a competing one. This leaves the option of increasing RNA specificity by NTR binding to RNA. There is a chemical shift perturbation in the NTR upon RNA binding. Again, backbone assignment, so the authors know which residues shift and mutational analysis is possible.

2. Figure 2: To me (as mentioned, not being an expert on this) all looks very similar and I have problems to be convinced about the differences in RNA binding specificity between the different constructs. There are no significance tests etc. in this figure. The BLI data is more convincing to me.

3. Figure 7: I do not see a significant difference between FL and Δ NTR in the invasion for sure and not so much in the viability either (panels C and D). Interestingly, between these two a significance indicator is not shown. Is the Kolmogorov-Smirnov test the correct one for significance (in panel B they use the more common unpaired t-test)? So maybe the NTR doesn't have a significant influence on function after all?

4. The affinities determined by BLI are very low. Even the NTR alone shows 1.2 μ M affinity, which would be in range of a single RRM domain, which would support a direct role of the NTR in RNA specificity. However, the NMR titration with RNA is not consistent with these affinity and if I understood it correctly, the same RNAs (25-mers) were used (P1). The NMR titration shows fast exchange (NTR, which would correspond to high micromolar affinity) and fast-to-intermediate exchange (La-module), which would correspond to mid-micromolar to low micromolar affinity (not low nanomolar). What is the problem here?

Minor points (but important to clarify/improve)

1. Figure 1 could be made clearer, which part in panel B is exactly the NTR, La-module, at this stage not clear if LaM means La-motif or La-module, the reader needs to find that out). So it would be good to improve panel B with adding domain boundary sequence numbers and possibly including information from panel C into this also.

2. The identified motifs in panel C of the NTR are R,K-rich (positively charged, so could bind RNA directly...or involved in LLPS?)

3. The authors mention an α/β -content estimation from CD data but this is not shown anywhere.

4. What about the CTR which (according to the authors) also influences LARP6's RNA binding specificity?

Reviewer #2

(Remarks to the Author)

Capraro, Abis, and colleagues present an original and thought-provoking body of work that provides deeper insights into LARP6 structure and function. Applying a thoughtful combination of omics and cell biology techniques, biophysical characterization, and computational prediction, they build a strong case for a previously unknown role of the intrinsically disordered N-terminal region of LARP6 in fine-tuning the specificity of La-RNA interactions through conformational restriction. I commend the authors on the rigorous execution of these studies and on the intuitive way in which they are presented. I support the publication of this manuscript, as both the biological insights and methodologies are valuable contributions to this research field.

Major Comments:

- While the case for the NTR's role in modulating LARP6 function is apparent, its specific relevance to LARP6's role in mesenchymal cancer invasiveness and viability is less clearcut. Although LARP6 knockdown unambiguously perturbed invasive capacity of U87MG cells in the collagen assay, the differential between the FL and Δ NTR LARP6 siRNA rescue was marginal. Further studies would be required to draw a definitive conclusion that it is the NTR's La-modulating function that drives this subtle rescue experiment difference, and not other effects of truncation on LARP6's localization, protein-protein interactions, etc. While I do not view this to be a barrier to publication, I encourage the authors to moderate their concluding statement in the Results section on the critical role of the LARP6 NTR in promoting cancer cell viability and invasion.

Minor comments:

- IP-OOPS methods: Please add specific details on proteolytic digest conditions, e.g. sample volumes, input protein concentrations (if known), protease:protein ratios, etc.
- LC-MS/MS methods: Possible typo for instrument model (Ultimate 3000 RSL nano HPLC -> Ultimate 3000 RSLC nano HPLC).
- IP-OOPS results: Consider including some brief commentary on whether there are any biases in proteolytic sites (and therefore peptide observability) between different regions of LARP6 that may influence the interpretation. Any knowledge on GFP-LARP6 peptide observability in the absence of UV-crosslinking may be helpful in this regard.

- Table 2: Recommend reformatting the table so that the KD values for La-module vs. NTD are side by side for each respective RNA oligo.
- ProteomeXchange/PRIDE deposition: Please add the protein FASTA used for the MaxQuant data processing to PXD064029, so that the precise LARP6 sequence used is archived.

Reviewer #3

(Remarks to the Author)

In this manuscript, Capraro et al show that the disordered N-terminal domain of the RNA-binding protein LARP6 fine-tunes RNA-binding specificity by limiting the conformational flexibility of the main RNA-binding domain, the La module. That IDRs tune the specificity and affinity of RNA-binding has been previously shown (PMID: 39353735) but the underlying molecular mechanism has not been investigated, to the best of my knowledge. Thus, this manuscript adds an important layer of information to the modulatory role of IDRs in RNA-binding. The function of RNA-binding by the IDR per se remains, however, unclear and the biological consequences of lacking an IDR for LARP6 seem subtle.

Specific comments:

- 1) The authors find that, in addition to binding RNA through the La module, LARP6 contacts RNA through the disordered N- and C- terminal regions. However, the role of RNA-binding through these additional regions in modulating the RNA-binding properties of LARP6 has not been investigated. The authors use complete deletions of the N-terminal IRD (up until the La motif), which is a region much broader than the concrete peptides that support RNA-binding (which in addition have not been precisely delineated). These large deletions have been useful to uncover a role of the N-terminal IDR in restricting the conformation of the La module, but they say little about the role of RNA binding by the IRD itself. This reviewer has the impression that similar conclusions could have been reached in this manuscript without the IP-OOPS data used to infer the RNA-binding regions of LARP6.
- 2) Does any IDR adjacent to an RBD have the same effect? Could, for example, the disordered region adjacent to the La module at the C-term, which does not bind RNA, have a similar effect than the N-terminal IDR in restricting RNA-binding selectivity? Please, notice that I am not referring to the newly identified RNA-binding region close to the SUZ-C domain, but to the region adjacent to the La module which has not been approached by deletion analysis.
- 3) Fine tuning of RNA-binding by the NTR does not seem to have a major biological impact in the in vitro cellular assays that the authors have chosen. The authors claim a difference in migration between FL and NTR (Fig 7C and lanes 1084-1086 of the main text), but the difference is not statistically significant. The authors should be careful not to overstate their findings. The only significant slight difference seems to rely on cell viability. Would injection of these cells in an animal model (e.g. mouse xenografts) results in a larger difference, for example in tumor growth? Such experiments would support the statement in the title, referring to 'pro-oncogenic activities of LARP6' which currently, in my opinion, are not sufficiently supported.
- 4) It is noticeable that the La module alone binds with higher affinity than the whole NTD. In other words, while the N-terminal IDR increases RNA-binding specificity, it also lowers the affinity of the La module for RNA. This should be discussed.
- 5) How do the RNA-binding properties of FL with NTR compare in vitro?
- 6) The work of Zigdon et al (PMID: 39353735) should be mentioned.

Other comments:

- Please, show the browser images for Figs 2C and 4A.
- Please, show a gel with the recombinant proteins used in Fig 4.

Version 1:

Reviewer comments:

Reviewer #1

(Remarks to the Author)

The additional NMR data (CSPs and relaxation) have really improved the manuscript, as they clearly show that the NTR interacts with the La-module with three sites on the NTR. And that this interaction is likely the reason of the rigidification of the La-module indicative from the SAXS data (to which I agree and I apologize for my confusion regarding this, possibly due to my initial confusion about what is what regarding the names of the constructs: LaM, La-Module, NTR, NTD...I think Figure 1 could still be improved)

However, in my opinion the reason for increased apparent RNA selectivity is still open for debate. The authors themselves have now added the following sentence in their discussion:

“Further to this, the disordered NTR may also contribute to the increased RNA binding selectivity by participating in direct contacts with RNA substrates, as well as modulating access to the La-module.”

What the data shows:

1. NTR rigidifies the La-module
2. NTR has direct interactions to the La-module and interacting sites on NTR are identified
3. NTR alone also binds RNA

4. Region S1 of NTR seems to be released from La-module upon RNA binding, but not S2 and S3

How I see it (hoping that I don't miss anything) is that how much each of the events contribute to RNA specificity is not clear.

The following things can contribute to RNA selectivity:

1. Rigidification of the La-module upon NTR binding
2. NTR contributes with direct RNA contacts to RNA specificity
3. Blocking of an RNA binding region on RRM and LaM by S1 (which would deselect weaker less-specific RNA targets)

They could all be responsible for increased RNA selectivity alone or altogether in any combination and weighting. To my understanding this has not been made clear by this work.

To 1., to show that this is an important contribution to RNA selectivity, the authors would need to know the cognate motif of the RRM and the LaM and then put different long spacers between both motif and measure affinity. Rigidified La-module would only bind strong to a certain motif-spaced RNA, while the flexible La-module could bind rather equally strong to differently motif-spaced RNAs. To my knowledge, all RNAs tested were 25 nts long and the separate cognate motifs of LaM and RRM are not known?

To 2., direct contacts are there (8 μ M or so affinity for NTR alone). I haven't found any CSPs for NTR to RNA (why not? Now easy to do?).

If NTR-RNA CSPs show which residues bind to RNA directly and if they are different from S1 then the three points could be nicely decoupled by mutational analysis and using different RNA lengths (although for the latter the motifs for RRM and LaM should be known).

I can agree to a certain extent that a mutational analysis can be difficult with such a complex system, especially if RNA binding and La-module binding by NTR cannot be nicely decoupled because of overlap (same residues on S1 bind RNA and La-module). So, I don't think it is necessary to do it for acceptance of this manuscript, but it would be great if the authors could unravel this in later work.

However, I think two things are necessary for acceptance:

1. Currently it sounds that the La-module rigidification is mostly responsible for RNA selectivity. This needs to be toned down. (it could actually be that this is not at all responsible for RNA specificity)
2. Addition of a "limitation of the study" section in which my points from above are addressed. This should list all the options of RNA specificity increase in an unbiased way and possible future experiments to decouple this (mutational analysis and testing different RNAs).

I hope I didn't miss anything and if so, I apologize in advance for the extra work and the delay this could have caused.

Reviewer #2

(Remarks to the Author)

Thank you to the authors for their thoughtful responses, manuscript adjustments, and additional data inclusions. The revised manuscript addresses all of my comments associated with the original submission, and I am supportive of the work's publication.

Reviewer #3

(Remarks to the Author)

The authors have convincingly addressed my concerns. The manuscript is an important contribution to the mechanisms by which disordered regions modulate RNA-binding selectivity.

Reply to reviewers

We thank the reviewers for the thorough and insightful review of the manuscript. We believe we have now addressed their concerns and implemented their suggestions to the best of our ability, as detailed below in our point-by-point rebuttal. Please note that for the ease of assessing the revisions, we have tracked the changes to the revised manuscript text in **red**.

Reviewer #1

The manuscript “an intrinsically disordered region mediates RNA-binding selectivity and pro-oncogenic activities of LARP6” by Capraro et al. describes a potential RNA-binding mechanism of LARP6 and how this mechanism influences its function in cancer cells. Using iCLIP and a mass-spectrometry based method to derive RNA specificity and identify the RNA binding region of LARP6, respectively, the authors speculate about the involvement of the IDR at the N-terminal region (NTR) in RNA specificity, as the data shows that deletion of the NTR decreases RNA specificity. They could demonstrate that this is true using BLI to derive affinities between protein constructs and a specific RNA 25-mer derived from one of the iCLIP hits. They already speculate here that the La-motif and the RRM (La-module) sample a large conformational space without the NTR and could therefore bind to more RNA sequences as compared to the full-length protein in which the NTR is thought to arrest the La-module in a certain conformation, which is specific for a certain RNA sequence. This they tried to validate with SAXS and NMR. Later they also attempted to show that a deletion of the NTR influences invasion and viability of cancer cells.

Disclaimer: the iCLIP and mass spectrometry data look sound and are partly well-validated with the in vitro data. However, I have not much hands-on experience on these experiments and the necessary statistics needed and can therefore not comment on this part of the manuscript.

While this story is well-written and appealing (and highly interesting, as there is still little known about RNA specificity beyond single classical RNA binding domains), the evidence for the presented mechanism is not convincing enough and it does not rule out other potential explanations (mechanisms) for the observations made. Before this manuscript can be considered at Nature Communications I recommend looking into the following major concerns I have:

Major concerns:

(Note: Point 1 contained several comments, so we split our replies to each part accordingly)

1- The mechanism, the authors presented, by which the NTR influences LARP6's RNA binding specificity is hard to imagine and not fully backed up by the data. Or in other words, other explanations could not be ruled out sufficiently. The authors claim that the La-module (La-motif + RRM) achieves higher RNA sequence specificity through conformational dynamics restriction imposed by the NTR. While I agree that the data shows that the NTR is needed for RNA specificity (BLI data and iCLIP, mass-spec), I am not convinced that it reduces the conformational sampling of the La-module and thereby increases RNA sequence selectivity.

> The reduction of the dynamics of the La-module in the presence of the NTR is one of the clearest evidence supported by our SEC-SAXS data (Figure 5e in revised manuscript, also reported below as Rebuttal Fig. 1), as shown by the overall reduction of the data points scattering of the NTD compared to the La-module in the Kratky plot analysis (recorded at the same synchrotron, in the same buffer, pH, temperature and protein concentration). As the La-module is formed by two structured domains, and in agreement with our previous published work (Lizarrondo et al, RNA biol. 2021¹), the SAXS Kratky plot of the La-module displays relative domain-domain dynamics that decrease when tethered to the NTR in the context of the NTD.

Please note that a later statement of the reviewer seems to agree with our interpretation: "I would agree that RNA rigidifies the independent tumbling of the La-motif and the RRM domain, as shown also for many tandem RRM domain proteins by SAXS and NMR [...] The NTR has a similar influence on the Kratky plot on the La-module without RNA present but not to such an extent. Comparing SAXS (Kratky and $P(r)$) of the La-module, NTR and NTD, it could be that the NTR binds transiently to the La-module, which would lead to a similar observation."

Rebuttal Figure 1: SEC-SAXS Kratky plot of NTD, NTR and La-module.

For single-stranded RNA, being very dynamic (maybe not in a full-length 3'UTR scenario but surely within a 25-mer), why should a rigid La-module be more sequence-specific than a mobile La-module (imagining that the La-motif tumbles independently of the RRM1)? If it would be about binding ssRNA within a certain RNA structure, I could imagine this to be a mechanism (e.g. Hollmann et al. 2020, Cell Reports), but with a 25-mer in vitro I doubt that.

> We respectfully disagree on this point with the reviewer. There are several published examples where a tandem domain becomes rigid upon ssRNA binding and this correlates with binding affinity/selectivity. One of such examples is an excellent study by Kang et al PNAS 2020 that the reviewer also kindly directed us to², where on page

7144 the authors write: “Overall dimensions and compactness of the U2AF2 RRM1,2 domains and their complexes with strong and weak Py-tract RNAs approximately correlate with their binding affinities, such that a more compact conformation is associated with a higher binding affinity (lower K_D ; Fig. 2b).”. We would like to highlight that the RNA oligos studied in this paper are also single stranded.

Another example, and closer to the system studied here, is the La-module of La: The LaM and RRM1 are independent in solution in the apo state and become rigid upon single stranded 3'oligoU interaction (Alfano et al., NSMB 2004; Kotik-Kogan Structure 2008; Lizarrondo RNA biol 2021^{1,3,4}).

[...] Other possibilities: *i)* the NTR is involved in RNA binding directly and increases specificity, *ii)* the NTR partly obscures the RNA binding region of the La-module and thus allows only certain RNA sequences to stay long enough to outcompete transient interactions of the NTR with the La-module (e.g. Kang HS et al., 2020, PNAS), the authors claim that this interaction is transient, which it seems to be indeed if there is an interaction (see minor point below). The evidence from NMR and SAXS is not conclusive. Upon RNA binding, the La-module and the NTD (NTR + La-module) are compacted as evident from SAXS, so I would agree that RNA rigidifies the independent tumbling of the La-motif and the RRM domain, as shown also for many tandem RRM domain proteins by SAXS and NMR (plenty of references). The NTR has a similar influence on the Kratky plot on the La-module without RNA present but not to such an extent. Comparing SAXS (Kratky and $P(r)$) of the La-module, NTR and NTD, it could be that the NTR binds transiently to the La-module, which would lead to a similar observation. Upon RNA binding, the NTR within the NTD could also bind to the RNA and thus increases compaction. The NMR data is also not conclusive. While the NTR is an IDR (the HetNOE data should be presented as a bar plot to show values between -1 and 1, HetNOE divided by reference, even without assignment, although assignment of the NTR should be straightforward, see below), the chemical shift perturbations (CSPs) between the NTR without La-module and with La-module are indeed very weak. The few CSPs could be in the connecting regions between NTR and La-module, which of course would shift when together in one construct compared to the separate ones (however, these shifts are likely larger). So why did the author not perform the backbone assignment of the NTR. This should be easily possible with a 250 μ M sample and this high-quality HSQC spectrum. Then the HetNOE data could be fully analyzed and the interaction with the La-module could be resolved at residue resolution (see below, mutational analysis). Similar for the La-module. The assignment is already available (as shown in the RNA titration experiment), why not identifying the interacting residues on the La-module side?

> In this section the reviewer asks us to better characterise the interaction between the NTR and the La-module to support the proposed molecular mechanism. In particular, the reviewer's concern is that our data, while supporting a conformational restriction mechanism, do not exclude additional mechanisms. Specifically, two other possible contributing mechanisms are proposed: 1- NTR and La-module bind RNA cooperatively, thus increasing selectivity, and/or 2- NTR is partially occupying the RNA binding region of the La-module and thus competing off certain unwanted RNA sequences.

We agree that this part needed further work and are grateful to the reviewer for highlighting the possibility of alternative mechanisms also at play. We followed the reviewer's suggestions to investigate this possibility in great detail, as detailed below:

- First, we obtained the NMR backbone assignment of the NTR which is now included in the new version of the manuscript. The assignment was performed on the NTR in isolation that gives high-quality NMR spectra and then transferred to the NTR in the context of the NTD (methods and Supplementary Figure 5 in the revised manuscript).
- We also repeated the entire NMR analysis at a higher field (950 MHz) and with the expert support of the MRC biomedical Centre at the Francis Crick institute, London. This led to a considerable improvement of the spectral quality, compared to the previously presented spectra of the NTD recorded at 800 MHz (Rebuttal Figure 2).

Rebuttal Figure 2: Comparison of data at 800 MHz reported in the original manuscript versus the data recorded at 950 MHz NMR, which is now included in the revised manuscript.

- Equipped with this information we can now perform an NTR-La module interaction analysis at residue resolution, following the reviewer's very useful suggestion. This enabled us to address the reviewer's comment of the CSPs that could have in principle come only from the connecting regions between NTR and La-module when tethered. As indicated in Rebuttal Figure 3 below (and Figure 5d in the revised manuscript), this

does not appear to be the case, as the large CSPs could be found in the regions approximately spanning residues 25-43 (termed S1) and 45-58 (termed S2), in addition to residues 77-83, which form the connecting region to the La-module (termed S3).

Rebuttal Figure 3: Chemical shift perturbation (CPS) values of the residues of the NTR in isolation vs. in the context of the NTD. R2 relaxation of the residues of the NTR in isolation (green) vs. in the context of the NTD (black). This figure is now reported as Figure 5d in the revised manuscript.

The CSPs are overall weak, and this is perfectly in agreement with transient fuzzy interaction with multiple averaged conformations co-existing at any time. To capture site-specific dynamic and transient behaviour of the NTR both in isolation and when tethered to the La-module, we performed R₂ relaxation analysis, which is better suited than the ¹⁵N-¹H Het-NOE analysis we previously presented (*Khan et al. Biophys J 2015*⁵). This shows an agreement with the CSPs, identifying regions that do interact with the tethered La-module, for which the relaxation rate is increased compared with that of the NTR alone (Rebuttal Fig. 3). This analysis has been included in the revised manuscript (Figure 5d).

- Having addressed these sets of the reviewer's comments, we then analysed by NMR the behaviour of the La-module and NTD in complex with RNA. These experiments were performed at 950 MHz at the Crick Institute. The R₂ relaxation experiments were performed in the presence of CTNNA1 P1 RNA to compare the dynamic properties of the NTR-in-the-NTD RNA-bound state versus apo. This revealed that the S2 and S3 regions do not experience large changes upon RNA binding, whilst R₂ values in S1 exhibited a considerable decrease consistent with increased intrinsic motion (Rebuttal Figure 4, which is Figure 6g in the revised manuscript). We reasoned that RNA binding would decrease the association of this part of the NTR with the La-module—and this is consistent with the negatively charged nature of residues comprising S1.

Rebuttal Figure 4: Changes in R_2 rates of NTR residues in isolation, in the context of NTD, or upon NTD-P1 RNA association, showing different dynamics behaviour of S1 and S2 segments. This figure is now reported as Figure 6g in the revised manuscript.

This interpretation is further supported by the CSP analysis comparing NTD in apo and P1 RNA bound state, as several residues in segment S1 revert (fully or partially) to their chemical shift values observed in the NTR in isolation, suggesting (at least partial) loss of interaction with La-module upon RNA binding (Rebuttal Figure 5 which is now Supplementary Figure 6f in the revised manuscript). Interestingly, the CSP analysis also shows that residues in the S2 segment experience distinctive perturbations, shifting to new positions. This may reflect either direct contacts with RNA, or conformational changes induced as a result of RNA binding. As this region overlaps with the IP-OOPS-identified potential RNA binding segment of NTR (residues 42-48; see Figure 1b of the revised manuscript), suggestive of the direct crosslinking to the RNA in living cells, the hypothesis of a direct RNA engagement is indeed most likely. The S2 segment does not have a particularly positively charged character, but does still contain an Arginine residue which may contribute to electrostatic interactions with the RNA backbone. It also contains a few aromatic residues that could contribute to association with RNA bases. This analysis has now been included in the revised manuscript (supplementary Figure 6f of the revised).

Rebuttal Figure 5: Delta CSP values between NTD/NTR and NTD/NTD+P1RNA in relation to R_2 relaxation rates (see text). The segment 2 (A45–E58) overlaps with the downstream region of the NTR IP-OOPS peptide (Fig 1b). This figure is now reported as Supplementary Figure 6f in the revised manuscript.

In conclusion, we are very grateful to the reviewer for their constructive suggestions, which have enabled us to refine the proposed mechanism by which the NTR modulates RNA binding selectivity in LARP6. **Our findings now support a composite mechanism, wherein RNA binding selectivity imparted by the NTR likely arises not only from reduced conformational sampling of the La-module but also from direct contacts of the NTR—specifically segment S2—with the RNA (a finding that also confirms our IP-OOPS data), and through modulating access to the La-module by segment S1.** This has now been incorporated into the revised manuscript's Results and Discussion sections (see manuscript lines 1081-1144).

Please note that having now performed an in-depth NMR analysis with the NTR assignment and with the entire NTD, we see that the data in the original Figures 6g, and 6h was indeed inconclusive as the reviewer pointed out, and we have therefore modified this part of the manuscript accordingly (see lines 1102-1110).

Then the authors could prepare point mutations on the NTR and La-module side to show that upon mutation the interaction between NTR and La-module is gone, and the RNA specificity should be also affected. Then the mechanism could be a competing one. This leaves the option of increasing RNA specificity by NTR binding to RNA. There is a chemical shift perturbation in the NTR upon RNA binding. Again, backbone assignment, so the authors know which residues shift and mutational analysis is possible.

> Whilst further point mutations to dissect the detailed contribution of the different sub-regions of the NTR towards its promotion of selective RNA binding is an exciting proposition, we believe this to be beyond the scope of our current manuscript, which aims to establish the domain/region-level mechanisms at play. The current version of this manuscript already integrates an extensive collection of data—spanning *in vivo* and *in vitro* systems—that together establishes our main findings regarding a novel role for NTR in promoting selective RNA binding by the La-module. These include (i) mass spectrometry-based mapping of LARP6–RNA interaction sites in living cells; (ii) extensive deletion mutagenesis combined with transcriptome-wide iCLIP to define the contributions of distinct LARP6 regions to RNA binding in living cells; (iii) computational (Pythia) as well as *in vitro* biophysical validation of the NTR's effect on promoting selective RNA binding; (iv) detailed NMR and SAXS-based analyses revealing the NTR mechanisms of action; and (v) functional assessment of NTR contribution towards cellular activities of LARP6, showing a significant role for this IDR region in mediating key activities of LARP6 in Glioblastoma cells . Notably, our NMR results already suggest sub-regional segments within the NTR that mediate RNA binding selectivity through different molecular mechanisms (i.e. S1 and S2), with our IP-OOPS data converging on the same S2 region as a further independent validation of its proposed function, setting the scene for future more focused studies onto these sub-regions.

In conclusion, while we fully agree that systematic mutational analysis of regions or individual residues (e.g. within S1 vs. S2) is the next logical step for further analyses, such work is not trivial and would require a considerable amount of time and resources

(including rigorous benchmarking of mutant proteins to rule out indirect confounding effects). We therefore believe that such work warrants a dedicated follow-up study, and respectfully suggest that the scope of the present manuscript remains focused on discovering, validating, and characterising the role of NTR in its entirety in modulating the RNA binding selectivity of LARP6.

2. Figure 2: To me (as mentioned, not being an expert on this) all looks very similar and I have problems to be convinced about the differences in RNA binding specificity between the different constructs. There are no significance tests etc. in this figure. The BLI data is more convincing to me.

> This is correct. Figure 2 shows that deletion of the NTR or CTR does not alter the overall repertoire of transcripts that interact with LARP6 in living cells (Figure 2e). The differences we report—shown later in Figure 3 (*in vivo*) and Figure 4 (*in vitro*)—occur at the binding-site level, where the NTR deletion leads to broader and less selective binding footprints on the targeted RNAs (e.g. Figures 3e, 4a, 4d, and 4e). This is because the La-module primarily drives RNA recognition (Figure 2b), while the NTR fine-tunes the local binding site selectivity.

3. Figure 7: I do not see a significant difference between FL and Δ NTR in the invasion for sure and not so much in the viability either (panels C and D). Interestingly, between these two a significance indicator is not shown. Is the Kolmogorov-Smirnov test the correct one for significance (in panel B they use the more common unpaired t-test)? So maybe the NTR doesn't have a significant influence on function after all?

> We are grateful for this comment. We have now done multiple repeats of this experiment and clearly see a significant difference between FL and Δ NTR in their ability to rescue invasion upon loss of endogenous LARP6 (see new Figure 7c). Regarding the statistical test, we were originally using the Kolmogorov-Smirnov test due to the fact that the original smaller datasets did not pass normality test (meaning data distribution could not be assumed to be normal). However, with the additional replicates, the datasets pass the normality test, so a common unpaired t-test is now used for significance calculation (see new manuscript Figure 7c).

With regards to cell viability, although the impact of endogenous LARP6 depletion on the viability of U87 cells is relatively mild (only a ~25% decrease after 72hrs of depletion, compared to ~50% decrease we reported previously using MDA-MB231 breast cancer cell – see *Dermitt et al., 2020⁶*), the FL construct, but not the Δ NTR can still rescue this effect, meaning that the NTR is needed for LARP6 function in this context as well.

4. The affinities determined by BLI are very low. Even the NTR alone shows 1.2 μ M affinity, which would be in range of a single RRM domain, which would support a direct role of the NTR in RNA specificity. However, the NMR titration with RNA is not consistent with these affinity and if I understood it correctly, the same RNAs (25-mers) were used (P1). The NMR titration shows fast exchange (NTR, which would correspond to high micromolar affinity) and fast-to-intermediate exchange (La-module), which would correspond to mid-micromolar to low micromolar affinity (not low nanomolar). What is the problem here?

> We agree with the reviewer that the behaviours observed in NMR experiments do not always align with thermodynamic parameters obtained with other biophysical techniques. This discrepancy has been observed by us and others in several systems and is likely attributable to aggregation effects at the concentrations typically employed in NMR experiments, altering thermodynamics and kinetic behaviours. In light of this, and in agreement with the reviewer's concerns, we conducted an orthogonal binding assay that independently confirms the findings from the BLI analysis (MST). These new data have been incorporated into the revised manuscript and are presented in Supplementary Figure 4g-l (see manuscript lines 910-913).

Minor points (but important to clarify/improve)

1. Figure 1 could be made clearer, which part in panel B is exactly the NTR, La-module, at this stage not clear if LaM means La-motif or La-module, the reader needs to find that out). So it would be good to improve panel B with adding domain boundary sequence numbers and possibly including information from panel C into this also.

> Great suggestion! We agree and have rectified this (residue numbers are now added to both Figure 1 panels b and c, and the abbreviations are also described in the associated figure legend).

2. The identified motifs in panel c of the NTR are R,K-rich (positively charged, so could bind RNA directly...or involved in LLPS?)

> We have addressed this in the analysis above. Based on our new NMR analysis, we believe the key region here is C-terminal of the identified crosslink-adjacent NTR peptide (see manuscript lines 1081-1100).

3. The authors mention an α/β -content estimation from CD data but this is not shown anywhere.

> We thank the reviewer for noting this. The α/β -content estimation comparing CD and NMR data for the structured La-module was originally included as another table in an earlier version of the manuscript, but later removed, as deemed of no particular added value. However, the text referring to the secondary structure estimation was inadvertently left in. We have now corrected this.

Please note that in the revised manuscript, we have included an estimation of the secondary structure propensity of the NTR using NMR (see revised Supplementary Fig. 5c-d and manuscript lines 965-967), which is most relevant to this story.

4. What about the CTR which (according to the authors) also influences LARP6's RNA binding specificity?

> We are currently working on the CTR as part of a different ongoing project, which investigates both its RNA-dependent and independent roles, so this will be the focus of our next work!

Reviewer #2 (Remarks to the Author)

Capraro, Abis, and colleagues present an original and thought-provoking body of work that provides deeper insights into LARP6 structure and function. Applying a thoughtful combination of omics and cell biology techniques, biophysical characterization, and computational prediction, they build a strong case for a previously unknown role of the intrinsically disordered N-terminal region of LARP6 in fine-tuning the specificity of La-RNA interactions through conformational restriction. I commend the authors on the rigorous execution of these studies and on the intuitive way in which they are presented. I support the publication of this manuscript, as both the biological insights and methodologies are valuable contributions to this research field.

Major Comments:

- While the case for the NTR's role in modulating LARP6 function is apparent, its specific relevance to LARP6's role in mesenchymal cancer invasiveness and viability is less clearcut. Although LARP6 knockdown unambiguously perturbed invasive capacity of U87MG cells in the collagen assay, the differential between the FL and Δ NTR LARP6 siRNA rescue was marginal. Further studies would be required to draw a definitive conclusion that it is the NTR's La-modulating function that drives this subtle rescue experiment difference, and not other effects of truncation on LARP6's localization, protein-protein interactions, etc. While I do not view this to be a barrier to publication, I encourage the authors to moderate their concluding statement in the Results section on the critical role of the LARP6 NTR in promoting cancer cell viability and invasion.

> We thank the reviewer for this comment. We have now performed additional repeats of the 3D invasion experiment, which now shows a clear and significant effect for the NTR (revised Figure 7c; see also response to the comment 3 of the reviewer #1). We do, however, agree that moderating the tone of the concluding statements is a good idea. This has now been done (see manuscript lines 1165 and 1189). We have also changed the title of our manuscript as per a related comment from reviewer #3, to avoid overextending our findings with regards to *in vivo* tumourigenesis.

Minor comments:

- *IP-OOPS methods:* Please add specific details on proteolytic digest conditions, e.g. sample volumes, input protein concentrations (if known), protease:protein ratios, etc.

> Done (see manuscript lines 172-189).

- *LC-MS/MS methods:* Possible typo for instrument model (Ultimate 3000 RSL nano HPLC -> Ultimate 3000 RSLC nano HPLC).

> Thank you for spotting the typo. This is now corrected (see manuscript line 195).

- *IP-OOPS results: Consider including some brief commentary on whether there are any biases in proteolytic sites (and therefore peptide observability) between different regions of LARP6 that may influence the interpretation. Any knowledge on GFP-LARP6 peptide observability in the absence of UV-crosslinking may be helpful in this regard.*

> Great suggestion! We had data on this but this was not originally included. We have now added this to the revised manuscript (see new Supplementary Figure 1b, and manuscript lines 682-686).

- *Table 2: Recommend reformatting the table so that the KD values for La-module vs. NTD are side by side for each respective RNA oligo.*

> Done (we now have rearranged the values so that they are sorted by oligo).

- *ProteomeXchange/PRIDE deposition: Please add the protein FASTA used for the MaxQuant data processing to PXD064029, so that the precise LARP6 sequence used is archived.*

> Unfortunately, we have not been able to modify our PRIDE entry but have instead provided a link to the FASTA sequence in our methods section (see manuscript line 199).

Reviewer #3 (Remarks to the Author):

In this manuscript, Capraro et al show that the disordered N-terminal domain of the RNA-binding protein LARP6 fine-tunes RNA-binding specificity by limiting the conformational flexibility of the main RNA-binding domain, the La module. That IDRs tune the specificity and affinity of RNA-binding has been previously shown (PMID: 39353735) but the underlying molecular mechanism has not been investigated, to the best of my knowledge. Thus, this manuscript adds an important layer of information to the modulatory role of IDRs in RNA-binding. The function of RNA-binding by the IDR per se remains, however, unclear and the biological consequences of lacking an IDR for LARP6 seem subtle.

Specific comments:

1) *The authors find that, in addition to binding RNA through the La module, LARP6 contacts RNA through the disordered N- and C- terminal regions. However, the role of RNA-binding through these additional regions in modulating the RNA-binding properties of LARP6 has not been investigated. The authors use complete deletions of the N-terminal IRD (up until the La motif), which is a region much broader than the concrete peptides that support RNA-binding (which in addition have not been precisely delineated). These large deletions have been useful to uncover a role of the N-terminal IDR in restricting the conformation of the La module, but they say little about the role of RNA binding by the IRD itself. This reviewer has the impression that similar conclusions could have been reached in this manuscript without the IP-OOPS data used to infer the RNA-binding regions of LARP6.*

> This is a fair criticism which we have now addressed in the revised manuscript. Our new NMR data now shows that the S2 segment of NTR (which nicely overlaps with our candidate IP-OOPS mapped crosslinking site) directly engages with the RNA. This occurs in addition to conformational restriction of the La-module, and likely further contributes to RNA selection. Accordingly, we now propose that the mechanism of NTR is one that is composite, involving also a direct cooperative binding to RNA (see manuscript lines 1081-1144; see also response to reviewer #1).

2) Does any IDR adjacent to an RBD have the same effect? Could, for example, the disordered region adjacent to the La module at the C-term, which does not bind RNA, have a similar effect than the N-terminal IDR in restricting RNA-binding selectivity? Please, notice that I am not referring to the newly identified RNA-binding region close to the SUZ-C domain, but to the region adjacent to the La module which has not been approached by deletion analysis.

> At the moment, we have no data on the C-term adjacent region of the La-module so cannot comment on this. Our focus on the NTR region was guided by our initial IP-OOPS experiment and the subsequent iCLIP datasets. Of course more work can be done on other regions of LARP6, but this will have to be the focus of future studies.

3) Fine tuning of RNA-binding by the NTR does not seem to have a major biological impact in the in vitro cellular assays that the authors have chosen. The authors claim a difference in migration between FL and DNTR (Fig 7C and lanes 1084-1086 of the main text), but the difference is not statistically significant. The authors should be careful not to overstate their findings. The only significant slight difference seems to rely on cell viability. Would injection of these cells in an animal model (e.g. mouse xenografts) results in a larger difference, for example in tumor growth? Such experiments would support the statement in the title, referring to 'pro-oncogenic activities of LARP6' which currently, in my opinion, are not sufficiently supported.

> As discussed in our response to reviewers #1 and #2, We have now done multiple additional repeats of our invasion assay experiment, and now clearly see a significant difference between FL and Δ NTR in their ability to rescue 3D invasion (see new Figure 7c). Hence, both pro-invasion and pro-proliferation effects of LARP6 require the function of NTR. However, we agree with the reviewer in moderating the tone of our concluding statements regarding these cell-based assays. This has now been done as outlined in response to reviewer #2. In addition, as per reviewer's suggestion, we have now also revised our manuscript title, removing the word 'pro-oncogenic', in order not to imply an overextension of our findings to *in vivo* tumourigenesis (see manuscript line 2).

4) It is noticeable that the La module alone binds with higher affinity than the whole NTD. In other words, while the N-terminal IDR increases RNA-binding specificity, it also lowers the affinity of the La module for RNA. This should be discussed.

> We thank the reviewer for this excellent suggestion. The additional NMR data that we now report in the revised manuscript clearly shows that the acidic segment of NTR (S1) binds to the La-module and needs to be displaced upon RNA binding (new Figure 6g; see also response to comment 1 of reviewer #1), thus explaining the observed

reduction in affinity of the NTD in comparison to La-module alone. We now discuss this in the revised manuscript (see manuscript lines 1081-1110 and 1236-1238).

5) How do the RNA-binding properties of FL with Δ NTR compare *in vitro*?

> A side-by-side *in vitro* comparison of full-length (FL) and the Δ NTR LARP6 mutant would have been conceptually ideal, but in practice, FL proved incompatible with our BLI assays, with significant aggregation preventing reliable kinetic/thermodynamic measurements. NMR is likewise not tractable at this size. We therefore had to focus on the biochemically tractable NTD, which isolates the La-module and its adjacent NTR, the two regions that our iCLIP results revealed as key to the RNA binding activity of LARP6. Importantly, we believe that the parallels we observe between our *in vivo* iCLIP and *in vitro* BLI results justifies this choice and support our conclusions without overextending them. For example, our BLI measurement on La-module vs. NTD show the same selectivity trends as in our iCLIP data with FL and Δ NTR (i.e. peak widening effects), indicating that the NTR-dependent tuning observed in our *in vitro* experiments is likely to be intrinsic to the NTD.

Rebuttal References

1. Lizarrondo, J., Dock-Bregeon, A.C., Martino, L. & Conte, M.R. Structural dynamics in the La-module of La-related proteins. *RNA Biol* **18**, 194-206 (2021).
2. Kang, H.S. et al. An autoinhibitory intramolecular interaction proof-reads RNA recognition by the essential splicing factor U2AF2. *Proc Natl Acad Sci U S A* **117**, 7140-7149 (2020).
3. Alfano, C. et al. Structural analysis of cooperative RNA binding by the La motif and central RRM domain of human La protein. *Nat Struct Mol Biol* **11**, 323-9 (2004).
4. Kotik-Kogan, O., Valentine, E.R., Sanfelice, D., Conte, M.R. & Curry, S. Structural analysis reveals conformational plasticity in the recognition of RNA 3' ends by the human La protein. *Structure* **16**, 852-62 (2008).
5. Khan, S.N. et al. Distribution of Pico- and Nanosecond Motions in Disordered Proteins from Nuclear Spin Relaxation. *Biophys J* **109**, 988-99 (2015).
6. Dermit, M. et al. Subcellular mRNA Localization Regulates Ribosome Biogenesis in Migrating Cells. *Dev Cell* **55**, 298-313 e10 (2020).

Reply to reviewers

We are pleased to hear that the reviewers are satisfied with our revisions, and we thank them for their thorough and insightful suggestions that significantly improved our study. Reviewers #2 and #3 had no further comments, and we have here addressed the last two remaining minor comments of reviewer #1:

Reviewer #1

.....However, I think two things are necessary for acceptance:

1. Currently it sounds that the La-module rigidification is mostly responsible for RNA selectivity. This needs to be toned down. (it could actually be that this is not at all responsible for RNA specificity)

- We respectfully disagree with the notion that our manuscript places more weight on the rigidification mechanism. In fact, in our abstract, results, figure legends, and discussion sections, we clearly and repeatedly state that a composite mechanism involving all three processes (rigidification, making additional contacts with RNA, and gating the RNA access to the La-module) is likely mediating the enhanced selectivity. **Please see revised manuscript lines 38-41, 100-103, 359, 415-419, 503-506, and 1488-1489.** We believe that all of these mechanisms are likely responsible so should be equally mentioned and highlighted.

2. Addition of a “limitation of the study” section in which my points from above are addressed. This should list all the options of RNA specificity increase in an unbiased way and possible future experiments to decouple this (mutational analysis and testing different RNAs).

- Thank you for this suggestion. We have now added a sentence to our discussion section highlighting this point. **Please see revised manuscript lines 487-488.**